# Despotism promotes dyadic cooperation through enhanced interdependencies in non-human primate societies

Cooperation is the cornerstone of human societies, and its emergence is firmly linked to enhanced tolerance and egalitarianism. However, evidence of profuse cooperation in less tolerant and despotic societies challenges this predominant view. The overarching interdependency hypothesis may resolve the conundrum. It posits that group-level interdependencies, like strength in numbers in colonially nesting species or allomaternal care in cooperatively breeding species, promote indiscriminate cooperation through enhanced tolerance. Crucially, this hypothesis also predicts that dyadic interdependence like friendships, nepotistic biases, or coalitions, selectively enhance tolerance, fostering discriminate cooperation in despotic species. Species belonging to *Macaca*, which have a similar social organization, yet remarkable variation in tolerance, hierarchy steepness, nepotistic biases, and coalitionary tendencies, provide an opportunity for testing the interdependency hypothesis. In social group settings, we experimentally study cooperation, prosociality, and tolerance in six macaque species spanning a tolerance gradient. Our findings reveal high dyadic cooperation in despotic societies, yet this cooperation is restricted to a few partners. Dyadic prosociality, kinship, and tolerance positively predict cooperation. Further, our agent-based models demonstrate that despotic societies have fewer but more stable bonds and, thus, higher dyadic interdependencies than in egalitarian societies. Our results suggest that interdependencies facilitate the emergence and maintenance of cooperation.

Cooperation – collaborative efforts of two or more individuals to achieve a common goal – helps maintain societies and provides benefits to group members[1–3]. Existing theoretical and empirical frameworks provide evolutionary explanations of animal cooperation between kin (through kin selection) and non-kin (through reciprocal altruism)[3,4]. Yet, what motivates individuals to cooperate and how they choose reliable partners during cooperation remains unclear at the proximate level. Such motivations are fundamental to the emergence and maintenance of cooperation[5]. Conventionally, group-level cooperation success has been linked to societies that exhibit enhanced social tolerance (cf. self-domestication hypothesis[6]) and prosociality (cf. cooperative breeding hypothesis[7]). Accumulating empirical

research also shows that individuals in several species that are neither self-domesticated nor cooperatively breeding exhibit prosocial and cooperative tendencies, primarily through reciprocity[8–16]. While reciprocity is important for various levels of cooperation[2,17,18], such as dyadic, group, and inter-group, it can be considered a positive behavioral expression of the structural interdependencies[19], defined by an individual's stake in another[20]. However, empirical evidence on how interdependencies may form, function, and eventually lead to cooperation remains limited. Building on the idea that interdependence has a considerable role in the evolution of human cooperation (cf. interdependence hypothesis)[20,21], an overarching interdependency hypothesis may explain what might cause tolerant and less tolerant

✉e-mail: bhattacharjee.debottam@gmail.com; edwin_van_leeuwen@eva.mpg.de; j.j.m.massen@uu.nl

species, and others in general, to show both group- and dyadic-level cooperative tendencies[13]. This hypothesis suggests that inter-dependencies at the group level, for example, strength in numbers in colonially nesting species or allomaternal care in cooperatively breeding species, can lead to enhanced within-group social tolerance and promote indiscriminate prosociality and cooperation. It further emphasizes that at the dyadic level, preferential strong associations or friendships, nepotistic biases, and reliance on coalitions may result in selectively enhanced dyadic tolerance, promoting discriminate pro-sociality and cooperation, particularly in less tolerant or despotic societies[12,13]. Studying the role of interdependencies in cooperation requires a comparative approach that includes closely related species with varying levels of social tolerance, nepotistic biases, and coali-tionary tendencies, but comparative empirical research remains scarce despite the potential to yield crucial insights into the evolution of cooperation.

Among group-living non-human primates, the genus *Macaca* consists of ~22 species that have similar social organizations and group compositions but strikingly different levels of group-level social tol-erance, dominance hierarchy steepness, and nepotistic biases[22]. This provides an excellent opportunity for comparative research to inves-tigate the role of interdependencies in promoting cooperation. Macaque societies, primarily focusing on adult females, show a suite of these phylogenetically (often) co-varying traits and are traditionally categorized into four grades of social styles (also known as tolerance grades) along a despotic–egalitarian gradient[23]. Typically, despotic societies (e.g., Grades 1 and 2) are considered to show steeper hier-archies, more frequent aggressive interactions, and lower group-level tolerance than egalitarian societies (e.g., Grades 3 and 4)[22,24–27]. How-ever, despotic societies show strong kin bonds and nepotistic biases, especially among females of the same matriline[22,23,25,28]. Non-kin strong bonds may also form in these societies, primarily through friendships and coalitions[12,29,30], benefiting individuals during (post-)conflicts and often helping to increase their social ranks. Such advantageous yet selective preferences among both kin- and non-kin group members

can indicate the presence of stronger dyadic interdependence in despotic than egalitarian societies, leading to potentially enhanced dyadic tolerance and discriminate prosocial- and cooperative-tendencies[13].

To test the predictions of the interdependency hypothesis spe-cifically with regard to dyadic cooperation, here we investigate 13 groups of macaques belonging to six species representing all four tolerance grades (Supplementary Table S1 and Fig. 1A). We conduct > 29,800 min of observations and implement three standardized experimental paradigms (Fig. 1B–D) in the macaques' existing social settings to empirically assess social relationships, tolerance ($n = 105$), and prosocial ($n = 96$) and cooperative tendencies ($n = 102$). While these experimental paradigms in the social settings allow all indivi-duals to freely participate, potentially capturing group-level patterns[15,31,32], the operational levels of our measures are explicitly dyadic. Together, this combination of observational and experimental approaches advance our knowledge on how dyadic cooperation might emerge from underlying interdependencies.

## Results
### Cooperation success and tolerance across macaque tolerance grades
Cooperation success was assessed from the loose-string experiment (Fig. 1B and Supplementary Data S4). Two self-trained individuals needed to pull two loose ends of a single string simultaneously to obtain food rewards without monopolization. We counted successful instances of cooperation out of 600 testing trials to calculate the dyadic cooperation success rate. Cooperation success ranged between 8.33% and 61.11% (mean ± standard deviation = 31.60% ± 18.75%). After controlling for group size effects, we found a negative correlation between cooperation success and macaque tolerance grades (Partial Bayesian correlation test (hereafter, correlation test): $r = -0.85$, $n = 9$, 89% credible interval or 89% crI = [− 0.96, − 0.54], Fig. 2A and Supple-mentary Table S2), suggesting that dyads in more despotic societies showed higher cooperation success than dyads in more egalitarian

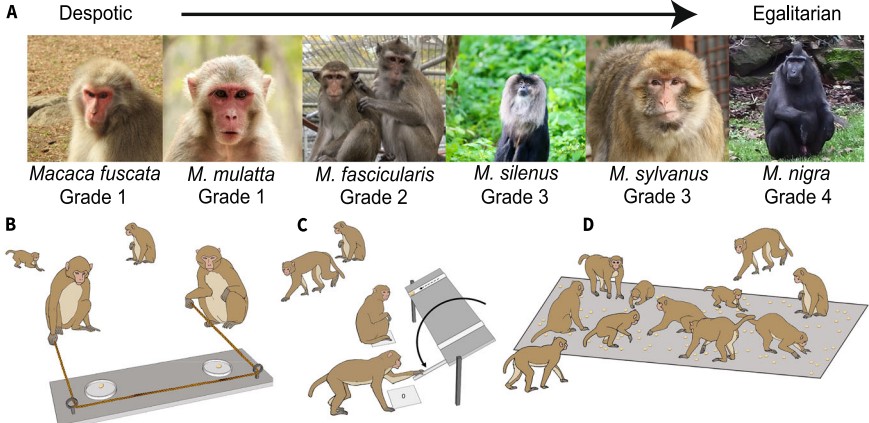

**A** Despotic → Egalitarian

*Macaca fuscata* Grade 1 | *M. mulatta* Grade 1 | *M. fascicularis* Grade 2 | *M. silenus* Grade 3 | *M. sylvanus* Grade 3 | *M. nigra* Grade 4

**Fig. 1 | Study species and the experimental paradigms. A** The six macaque species representing four tolerance grades – *Macaca fuscata* (Japanese), *M. mulatta* (Rhesus), *M. fascicularis* (Long-tailed), *M. silenus* (Lion-tailed), *M. sylvanus* (Barb-ary), and *M. nigra* (Crested). **B** We tested macaques' tendencies to cooperate using a loose-string paradigm (10 groups and 102 individuals, Supplementary Data S1). Two self-trained individuals needed to pull two loose ends of a single string simultaneously to obtain food rewards (Supplementary Movie S1). The string unthreaded when only one individual pulled, i.e., during uncoordinated pulling. **C** We used a group service paradigm to identify individuals with proactive prosocial preferences (9 groups and 96 individuals, Supplementary Data S2). Individuals could proactively provide food to group members (Supplementary Movie S2) by understanding the contingencies of two control (empty and blocked) conditions. We tested the tolerance of group members who participated in the group service experiment using a food distribution assessment phase (Supplementary Movie S3). **D** We conducted a co-feeding tolerance test to assess the group-level tolerance of the macaque groups (11 groups and 105 individuals, Supplementary Data S3). Contingent on group size, we created a plot of a particular size where a pre-determined number of peanuts were distributed (Supplementary Movie S4). The cumulative proportion of a group in the plot at regular intervals or scans during eight experimental sessions provided information about their tolerance levels. Due to COVID-19 restrictions, uniform data across all study groups could not be col-lected, leading to variations in sample sizes in analyses (Supplementary Table S1, S2 and Supplementary Data S1–S3). Macaque photo credits (A): Jana Jäckels, Jorg Massen, Gaia Zoo, and Debottam Bhattacharjee. Illustration of experimental para-digms (B-D): Veera Schroderus.

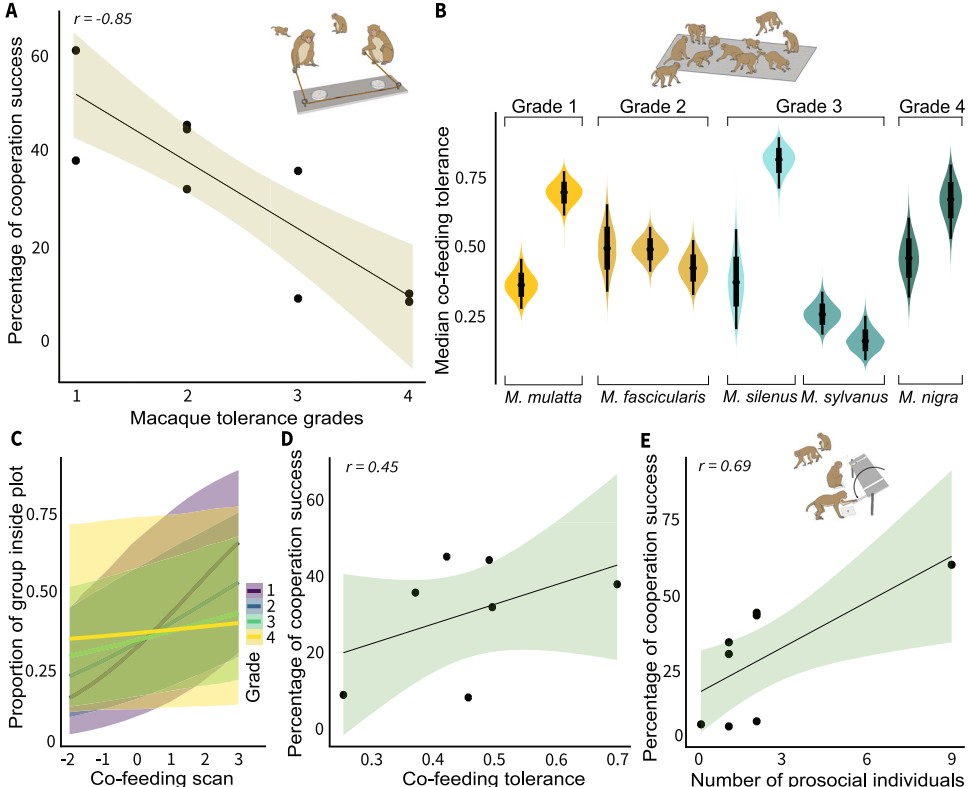

**Fig. 2 | Cooperation success and its associations with tolerance and prosociality across macaque tolerance grades. A** Cooperation success derived from dyads in different macaque species categorized under four tolerance grades. A negative relationship (Partial Bayesian correlation: $r = -0.85$, $n = 9$ groups, 89% crl = [−0.96, −0.54]) indicates that more despotic societies had higher cooperation success than egalitarian societies. Dots represent macaque groups; the line represents posterior correlation; the shaded band represents 89% crl. **B** Group-level co-feeding tolerance across societies belonging to five species and four tolerance grades, estimated using a Bayesian mixed-effects model fitted to scan-level data (87 sessions), with group nested within species as random effects ($n = 11$ groups). Violin plots represent posterior distributions of tolerance estimates; width reflects posterior density; black points indicate posterior median values; vertical extent represents 89% crl. **C** Posterior effects of the interaction between tolerance grade and co-feeding scan number on the proportion of members present in the plot. Predictions are based on a Bayesian mixed-effects model fitted to scan-level data ($n = 11$ groups, 537 scans from 87 sessions), with sessions nested within groups and groups within species as random effects. Lines represent posterior predicted proportions; shaded bands represent 89% crl. **D** Relationship between co-feeding tolerance and cooperation success. A weak to moderate positive association was found ($r = 0.45$, $n = 7$ groups, 89% crl = [− 0.30, 0.86]). Dots represent macaque groups; the line represents posterior correlation; the shaded band represents 89% crl. **E** Relationship between the number of individuals with prosocial motivations and cooperation success. A positive association was found ($r = 0.69$, $n = 8$ groups, 89% crl = [0.14, 0.92]). Dots represent macaque groups; the line represents posterior correlation; the shaded band represents 89% crl. Macaque illustration credits (**A, B, E**): Veera Schroderus. Source data are provided as a Source Data file.

societies. To understand the causes, we first tested whether the macaque co-variation framework assumption regarding the steepness of hierarchies (i.e., steeper hierarchies in more despotic societies) is met (cf. Fig. 1A).

A negative but weak correlation was found between tolerance grades and hierarchy steepness ($r = -0.33$, $n = 12$, 89% crl = [− 0.70, 0.19], Supplementary Data S4 and Supplementary Table S2), indicating a trend that despotic societies indeed have steeper hierarchies than egalitarian ones. However, the assumption of the co-variation framework is built upon within-group interactions among adult females[22,23,25]. Accordingly, based on only adult females in the groups (applicable to 10 groups), we recalculated steepness and re-investigated the relationship. In these data, a stronger negative correlation was found ($r = -0.61$, $n = 10$, 89% crl = [− 0.87, − 0.11], Supplementary Data S4 and Supplementary Table S2), reliably indicating that the assumption was clearly met. However, we concentrated on the full groups with females and males in the remainder of the analyses to have enhanced statistical power. This also allowed all sex combinations to be included in investigations at the dyadic level, as between-sex bonds can be important in primates[33,34]. In addition, as the dyadic relationships included interactions among all group members and given that

our experimental paradigms were employed in social settings, we conducted analyses using both sexes and age classes. Next, we investigated tolerance and prosociality and examined their associations with cooperation success.

Social tolerance is a central concept in primatology, where enhanced tolerance is positively linked to cooperation[6,7,35–37]. Yet, varying definitions and methodologies are present concerning its structural and behavioral constructs[38]. We assessed within-group social tolerance using a standardized co-feeding peanut plot experiment[39–41] (cf. Fig. 1D and Supplementary Data S3). By contrast, dyadic tolerance was assessed from a social tolerance phase of the cooperative loose-string experiment (Fig. 1D). We used 'co-feeding tolerance' to denote our experimentally obtained within-group social tolerance. Substantial intra- and inter-species variation in co-feeding tolerance was found (Fig. 2B). In general, however, and contrary to the co-variation framework's assumption, more despotic societies showed a higher co-feeding tolerance tendency than the egalitarian ones did (Fig. 2C). We further conducted a correlation test between the co-feeding tolerance and Pielou's evenness index (or Pielou's $J$[42]), calculated from a food distribution assessment phase of the group service experiment (cf. Fig. 1C). A strong positive relationship was found

between co-feeding tolerance and Pielou's *J'* (*r* = 0.92, *n* = 7, 89% crl = [0.59, 0.99], Supplementary Table S2), validating our co-feeding tolerance measure in capturing macaques' within-group social tolerance. A low to moderate positive relationship was found between within-group co-feeding tolerance and dyadic cooperation success (*r* = 0.45, *n* = 7, 89% crl = [− 0.30, 0.86], Fig. 2D and Supplementary Table S2). In addition, a positive correlation was found between cooperation success and the number of individuals with prosocial motivations (or 'prosocial' individuals) those groups had (*r* = 0.69, *n* = 8, 89% crl = [0.14, 0.92], Fig. 2E, Supplementary Data S2 and Supplementary Table S2), corroborating the positive relationship between prosociality and cooperation[7,36]. We performed this analysis by excluding the group with nine individuals with prosocial motivations to eliminate the potential bias of an outlier. The revised r was found to be 0.50, thus still suggesting a positive relationship between prosociality and cooperation.

## Predictors of dyadic cooperation

Theoretically, each self-trained participating individual within a group had the opportunity to cooperate with other participating members (cf. Supplementary Data S1). However, cooperation did not occur among all dyads, resulting in a zero-inflated dataset. Subsequently, we used a sequential decision-making hurdle approach to analyze our zero-inflated cooperation data. In this approach, we first investigated the probability or likelihood of cooperation (first hurdle: yes or no, i.e., whether a dyad successfully cooperated at all), followed by the intensity or magnitude of cooperation (second hurdle: dyadic cooperation success ≥ 1).

Of all potential dyads in Grade 1 to 4 societies, 31.9%, 36.9%, 46.1%, and 88.8% cooperated at least once, respectively (Fig. 3A–D and Supplementary Data S5). To understand how the magnitude of cooperation (cooperation success ≥ 1) was distributed, i.e., whether fewer dyads cooperated more or all dyads cooperated uniformly, we performed normality tests (Supplementary Data S5). We further used bootstrapping analysis using 10,000 resamples[43] to control for the sample size variations (Fig. 3E–H). These results confirmed that the observed patterns were not influenced by sample size differences. More despotic societies (grades 1 and 2) showed positively skewed patterns, whereas normal patterns were observed in more egalitarian societies (grades 3 and 4) (Fig. 3E–H). The overall low percentages of cooperative dyads and the skewed distribution of magnitude suggest a higher dyad specificity, thus partner choice, in despotic than in egalitarian societies.

Due to the unavailability of uniform data across groups (Supplementary Table S1 and Supplementary Data S1–S4), two sets of models were built for analyzing dyadic cooperation (Supplementary Table S3–6). In the first set of models, we found evidence of prosociality, dyadic social tolerance, and kinship positively predicting the likelihood of cooperation (Fig. 4A and Supplementary Table S3). Prosocial dyads were more likely to cooperate than non-prosocial dyads (Est = 1.41, 89% crl = [0.58, 2.27], *n* = 214, pd = 0.99). Dyads comprised of individuals with higher social tolerance were more likely to cooperate than dyads with lower tolerance (Est = 0.86, 89% crl = [0.40, 1.37], *n* = 214, pd = 0.99). Dyads with kin members were more likely to cooperate than dyads consisting of non-kin members (Est = 1.23, 89% crl = [0.29, 2.18], *n* = 214, pd = 0.98). Notably, kinship only explained 35% of the data variance. Unlike in analyses on likelihood, only a positive effect of dyadic social tolerance was found in predicting the magnitude of cooperation (Fig. 4B and Supplementary Table S4). Dyads comprised of individuals with higher social tolerance cooperated more than dyads with lower tolerance (Est = 0.23, 89% crl = [0.09, 0.40], *n* = 80, pd = 0.99). In the second set of models, except for a negative effect of group size on the likelihood (Est = − 0.26, 89% crl = [− 0.48, − 0.07], *n* = 115, pd = 0.98), no effects of tolerance grades,

grooming, and aggression indices were found on the likelihood and magnitude of cooperation (Fig. 4C, D and Supplementary Table S5, 6).

In an additional set of models, focusing on the results of the group-service prosociality experiment, we found that dyadic prosocial food provisioning was predicted by kinship and dyadic social tolerance (Supplementary Fig. S1, Supplementary Data S3 and Supplementary Table S7, 8). Kinship positively predicted both the likelihood (Est = 1.13, 89% crl = [0.33, 1.96], *n* = 302, pd = 0.99) and magnitude (Est = 0.98, 89% crl = [0.45, 1.51], *n* = 49, pd = 0.99) of prosocial food provisioning, whereas dyadic tolerance only positively predicted the likelihood (Est = 0.98, 89% crl = [0.57, 1.53], *n* = 302, pd = 1).

Our findings corroborate previous results that prosociality, social tolerance, and kinship are key proximate mechanisms of cooperation[1,3,7–9,36,44]. Yet, most importantly, we emphasize that cooperative dyads in more despotic societies are fewer and have higher selectivity in comparison to those in more egalitarian societies. We also expected these strong dyadic interdependencies in despotic societies to be reflected in grooming interactions (cf. Fig. 4C, D). Therefore, we conducted additional analyses to understand the grooming patterns, where exchange of commodities can be observed[18]. Based on the frequency of grooming, we built social networks and calculated global transitivity, reciprocity, and modularity. Transitivity provides information on the level of clustering and is particularly useful for interpreting weighted and directed social networks[26,45]. Reciprocity, by contrast, is a measure showing the difference between grooming efforts given and received. Modularity denotes how a network can be divided into communities where individuals groom specific partners more frequently than chance level[27]. We found both transitivity and reciprocity to be positively correlated with macaque tolerance grades (Transitivity: *r* = 0.73, *n* = 12, 89% crl = [0.38, 0.90]; Reciprocity: *r* = 0.70, *n* = 12, 89% crl = [0.33, 0.89], Fig. 4E and Supplementary Table S2), indicating the presence of more transitive and reciprocal grooming networks in more egalitarian societies. Further, more egalitarian societies showed a lower tendency of grooming modularity than more despotic societies (*r* = − 0.36, *n* = 12, 89% crl = [− 0.72, 0.16], Supplementary Table S2). The observed grooming patterns thus aligned with the assumptions of the co-variation framework[22,23,25]. Accordingly, these findings opened up the idea that more despotic societies possibly restrict the number of strong (cooperative) bonds based on affiliative interactions (e.g., grooming, also see refs. 16,18,46 for food sharing) and that they may be used as a commodity traded in exchange for alternate services, like support during conflicts (*sensu* biological market theory)[47]. As food sharing is not common in macaque societies, we constructed agent-based models simulating macaque social behavior primarily based on grooming interactions and emotional bookkeeping (see e.g., ref. 48,) along a despotic-egalitarian gradient.

## Emergence of strong social bonds

Strong social bonds can promote cooperation[37,49]. To investigate how dyadic bonds may emerge in despotic and egalitarian societies, we employed agent-based EMO-models[50–52]. EMO-models simulate macaque social behavior, where 'agents' or individuals within a group are capable of emotional bookkeeping[17,53], integrating information of past affiliative interactions into a partner-specific LIKE attitude value. These LIKE relationships range between '0' and '1', with higher values indicating stronger bonds. To test the effect of despotism vs egalitarianism on LIKE relationships, we incorporated a Hierarchy steepness variable. Despotic, intermediate, and egalitarian societies were assigned steepness values of '1', '0.6', and '0.2', respectively. The effect of steepness on LIKE relationships was investigated in diverse environmental conditions: three increase speeds (i.e., the speed with which LIKE attitude increases when grooming is received: fast, intermediate, and slow), three decrease speeds (i.e., the speed with which LIKE attitude decreases in the absence of being groomed; fast,

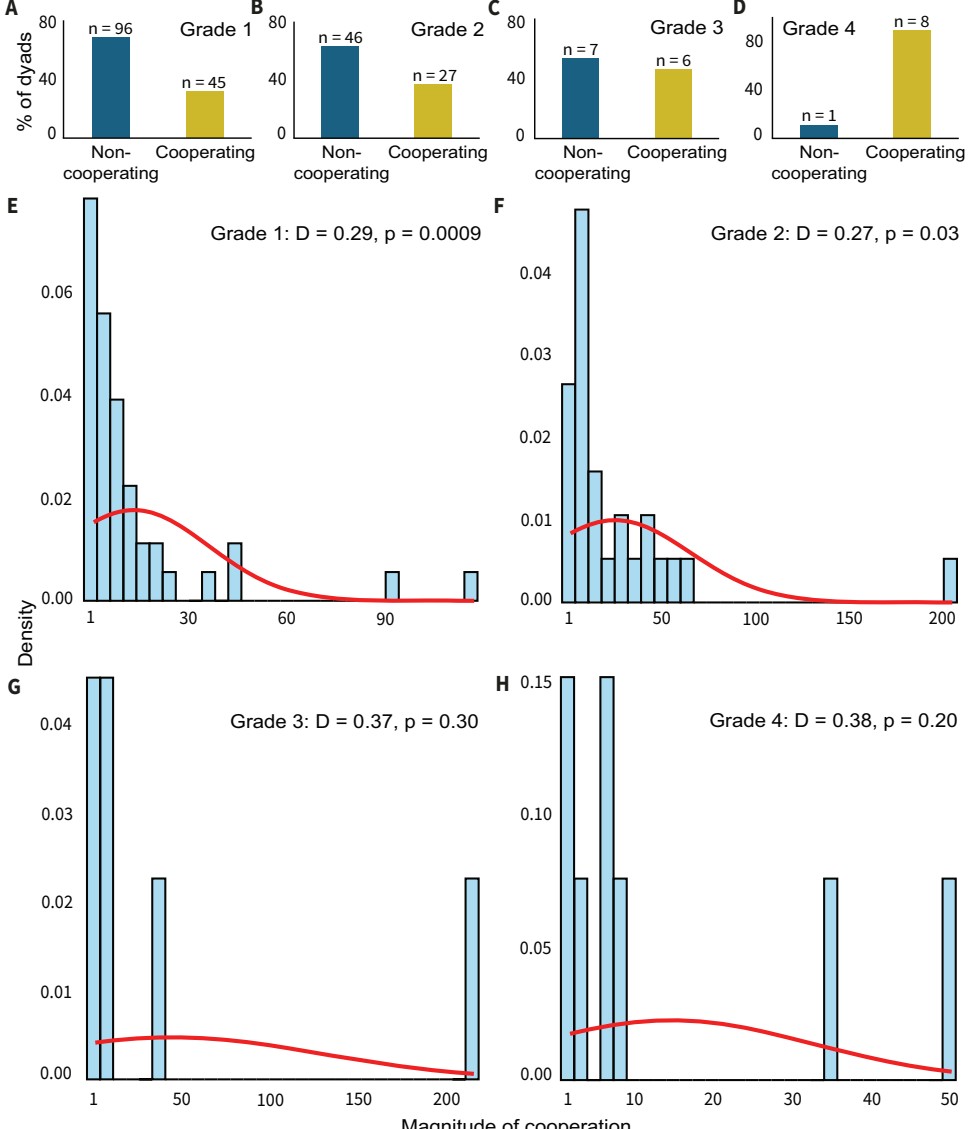

**Fig. 3 | Dyad specificity in cooperation across macaque tolerance grades.**
**A–D** The percentages of cooperating and non-cooperating dyads in Grade 1, Grade 2, Grade 3, and Grade 4 societies. Note that the total number of potential dyads (i.e., both non-cooperating and cooperating) was calculated based on only self-trained individuals. No inferential statistical tests were performed. **E–H** The distribution of magnitude values (cooperation success ≥ 1) in Grade 1, Grade 2, Grade 3, and Grade 4 societies with a histogram and an overlaid distribution curve (Two sided Kolmogorov-Smirnov tests for normality with D statistic and *p*-values). The histograms (in sky blue) represent the empirical distribution of the data, with the y-axes displaying the density of the magnitude of cooperation. The red lines show the theoretical normal distribution based on the sample mean and standard deviation. Source data are provided as a Source Data file.

intermediate, and slow), and two LIKE dynamics ('easy-going dynamics' and 'picky dynamics'[54]). In picky dynamics, it is more difficult to establish new relationships, but once a strong relationship is formed, the quality decreases more slowly over time compared to a relationship of the same quality in easy-going dynamics. We also assessed the stability of LIKE relationships over time[54]. We investigated LIKE relationships at five different time points and calculated a correlation score between consecutive time points. The resulting four scores were averaged for each simulation to obtain a 'stability' score.

In line with our expectation, the number of potential interaction partners and the emergence of high LIKE relationships were restricted by the steep dominance hierarchy (values of '1' and less so in '0.6') in despotic societies (Fig. 5A–C and Supplementary Fig. S2–6). Strong bonds formed almost exclusively among individuals relatively close in their dominance ranks in more despotic societies. Notably, in matrilineal macaque societies, close rank relationships are often restricted to kin members. The pattern was consistent in situations of all decrease speeds. However, slower decrease speeds resulted in a higher absolute number of high LIKE relationships overall. In comparison to despotic and intermediate hierarchical societies, the dominance hierarchy had little influence on the emergence of high LIKE relationships in egalitarian societies (value of 0.2). When calculating the stability of the LIKE relationships, we found that they were more stable in despotic than egalitarian societies (Fig. 5D).

## Discussion

Experimental evidence on the motivational mechanisms facilitating spontaneous dyadic cooperation was lacking, especially in less-tolerant societies[55]. We show that dyads in more despotic societies exhibit higher cooperation success than those in egalitarian societies. Although evidence of a weak positive correlation between within-

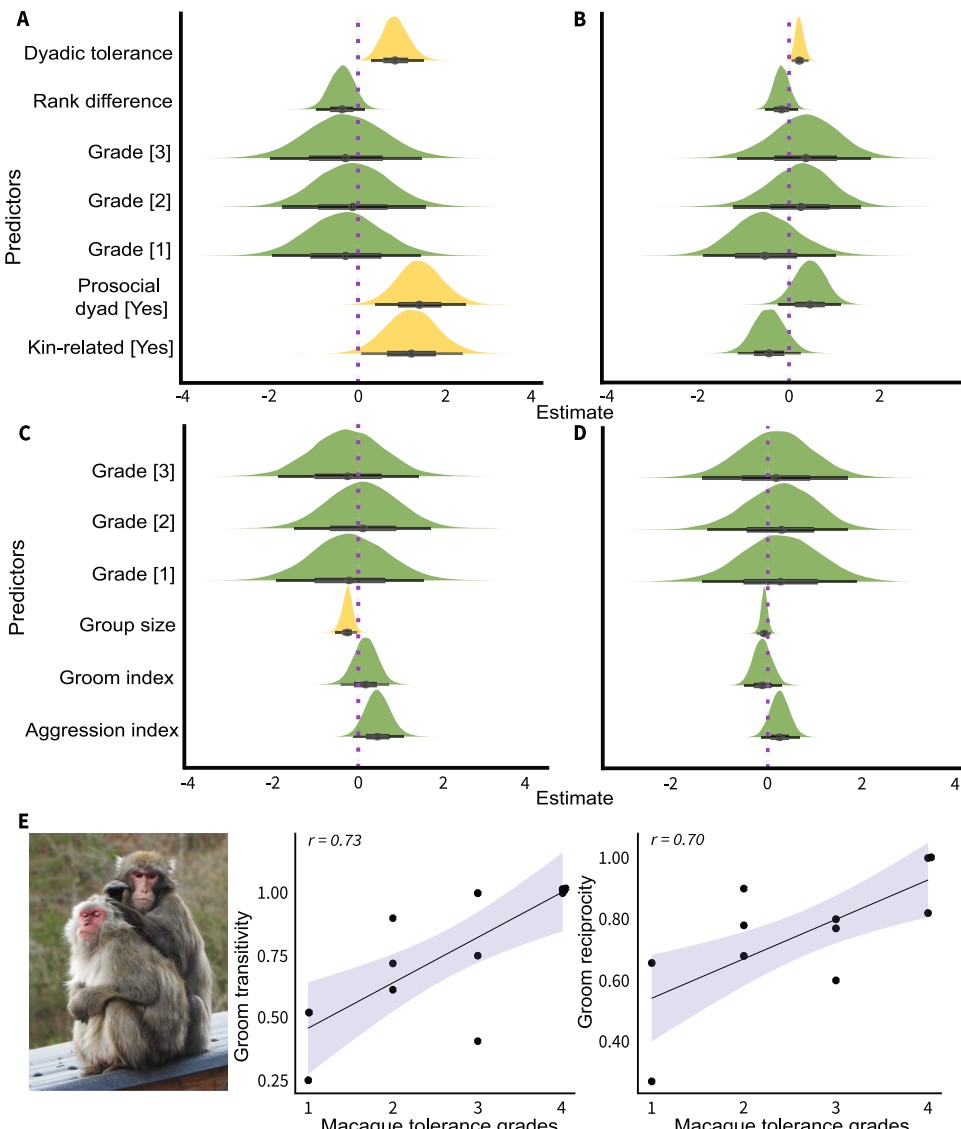

**Fig. 4 | Dyadic cooperation and grooming patterns. A** Posterior effects of dyadic social tolerance, rank difference, tolerance grades, prosociality, and kinship on cooperation likelihood, estimated from a Bayesian model (*n* = 214 dyads). Only strong effects of prosocial dyads (Est = 1.41, 89% crl = [0.58, 2.27]), social tolerance (Est = 0.86, 89% crl = [0.40, 1.37]), and kin-relation (Est = 1.23, 89% crl = [0.29, 2.18]) were found. **B** Effects of dyadic social tolerance, rank difference, tolerance grades, prosociality, and kinship on cooperation magnitude, estimated from a Bayesian model (*n* = 80 dyads). Only a strong effect of dyadic social tolerance (Est = 0.23, 89% crl = [0.09, 0.40]) was found. **C** Effects of aggression, grooming, group size, and tolerance grades on the likelihood of cooperation, estimated from a Bayesian model (*n* = 115 dyads). Only group size effect was found (Est = − 0.26, 89% crl = [− 0.48, −0.07]). **D** Effects of aggression, grooming, group size, and tolerance grades on the magnitude of cooperation, estimated from a Bayesian model (*n* = 47 dyads). No strong associations were found. In (**A**–**D**), yellow colors indicate strong effects; the width of the 'half-eye' represents data distribution (89% crl); solid black points on horizontal bars indicate median values; vertical purple dashed lines indicate a parameter estimate of zero, i.e., overlap of the crl with this line suggests no effects. **E** Relationships between macaque tolerance grades and grooming transitivity and reciprocity (*n* = 12). The positive associations indicate higher transitivity (Partial Bayesian correlation: *r* = 0.73, 89% crl = [0.38, 0.90]) and reciprocity (*r* = 0.70, 89% crl = [0.33, 0.89]) in more egalitarian societies. Dots represent individual groups; solid lines represent posterior correlation; shaded bands represent 89% crl. Macaque photo credit (**E**): Jana Jäckels. Source data are provided as a Source Data file.

group co-feeding tolerance and dyadic cooperation at the group level was found, surprisingly, this co-feeding tolerance did not align with the attributed social tolerance grades of macaque societies. Such a con-trasting pattern could be present due to including individuals of both sexes, as well as juveniles, as opposed to social tolerance constructs that are built exclusively on interactions among adult females[23,25]. Crucially, dyadic social tolerance was the best predictor of dyadic cooperation, along with partial positive effects of prosociality and kinship. Our results, therefore, suggest selectively enhanced dyadic tolerance among individuals in more despotic societies, promoting dyadic cooperation.

Species-level differences of co-varying traits in macaques along the despotic-egalitarian gradient are not always continuous[27,56]. Simi-larly, in group-level dominance hierarchy steepness and grooming metrics, we found considerable overlaps across tolerance grades. For instance, some groups classified as more egalitarian (i.e., grades 3 and 4) exhibited hierarchy steepness values comparable to those observed in more despotic groups (i.e., grades 1 and 2), with similar overlaps also apparent in grooming behaviors. These patterns may indicate group-level signatures or group-specific ways in which non-human primates behave[8,40]. Yet, despite the group-level variations, we did find strong associations between tolerance grades with hierarchy steepness and

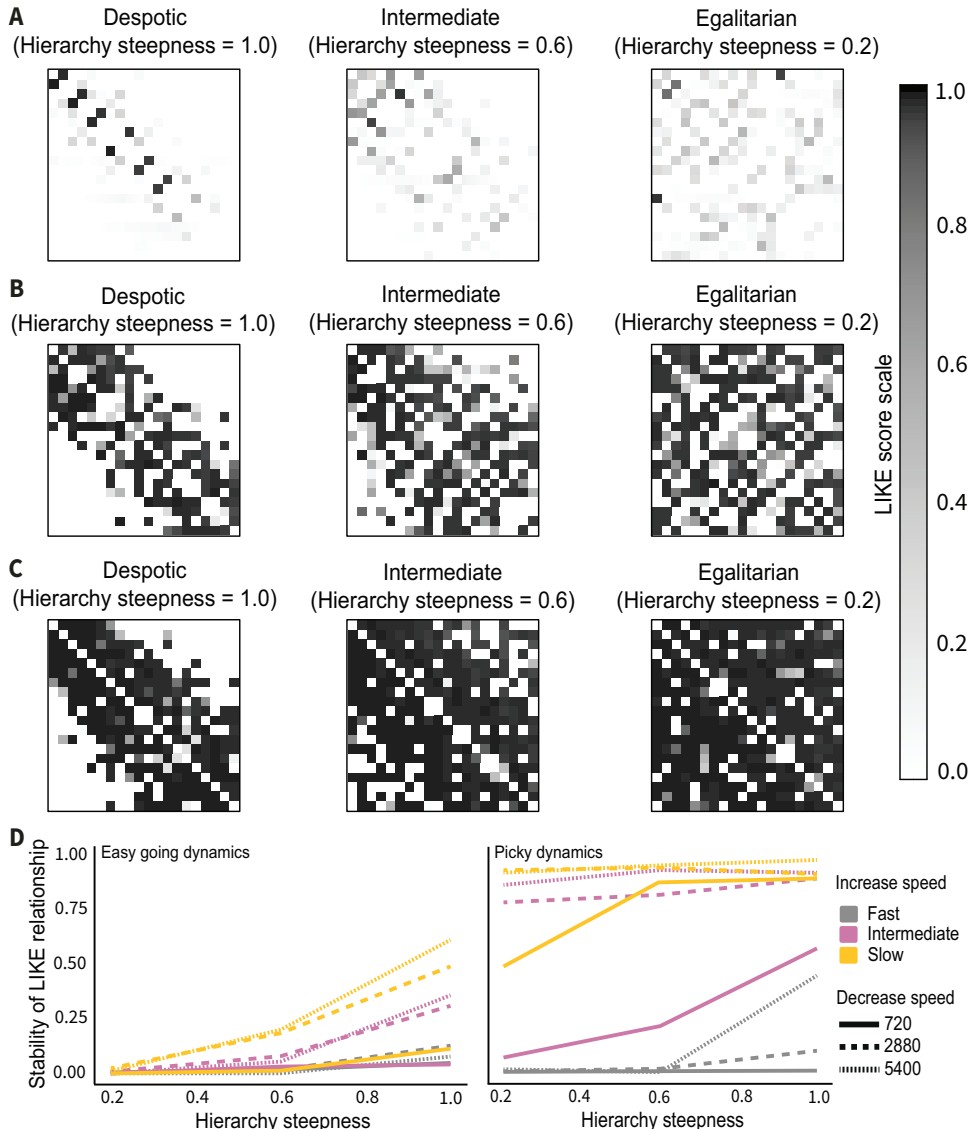

**Fig. 5 | Agent-based EMO-models and the emergence and stability of LIKE relationships in societies along a despotic-egalitarian gradient. A** LIKE relationships in societies with an intermediate increase speed, fast decrease speed, and picky dynamics. **B** LIKE relationships in societies with an intermediate increase speed, intermediate decrease speed, and picky dynamics. **C** LIKE relationships in societies with an intermediate increase speed, slow decrease speed, and picky dynamics. In (**A**−**C**) on the y-axes, individuals are ordered from low ranking (top row) to high ranking (bottom row), and on the x-axes, from low ranking (left) to high ranking (right). Each square represents a LIKE attitude from one individual to another, indicating their LIKE relationship. A LIKE relationship ranges from 0 (white) to 1 (black), with higher values indicating stronger bonds. **D** The stability of LIKE relationships in despotic (Hierarchy steepness = 1.0), intermediate (Hierarchy steepness = 0.6), and egalitarian (Hierarchy steepness = 0.2) societies using the easy-going and picky dynamics. Only one run is shown for all parameter settings. Repeats of runs using the same settings but a different random seed always resulted in very similar stability scores, except for runs with picky dynamics, a slow increase speed, and a fast decrease speed. See Supplementary Fig. S2–6 for EMO-model simulation runs with fast and slow increase speeds and easy-going dynamics. These simulations showed either similar results or very stable LIKE relationships across the whole population, regardless of Hierarchy steepness (slow and intermediate increase and decrease speeds) or almost no LIKE relationships, regardless of the Hierarchy steepness (fast decrease and slow increase). As the latter two are not what we know from empirical data, these seem ecologically less relevant. Source data are provided as a Source Data file.

grooming. Importantly, those associations were consistent with the prediction of the macaque co-variation framework. Nevertheless, these findings may also suggest that the despotic-egalitarian classification captures only broad tendencies in social style and thus should be applied within a comparative framework.

In line with the prediction of the interdependency hypothesis with regard to dyadic cooperation, the percentages of cooperating dyads in despotic societies were lower, and higher cooperation success was even more restricted to specific dyads than in egalitarian societies. Agent-based models consistently showed that steeper hierarchies of more despotic societies allow for the emergence of fewer but more stable social bonds along the dominance hierarchy. Thus, steep hierarchies seem to force individuals into strong and interdependent bonds with those close in rank. However, based on our empirical data, the smaller rank differences only showed a tendency to promote cooperation positively (but see ref. 12). We suspect this overall effect might have been weakened by the data of more egalitarian societies that are not so restrictive regarding partner choice while cooperating. This could also indicate that successful cooperative dyads in despotic societies do not exclusively rely on frequent reciprocal affiliative exchanges and instead may use 'Machiavellian' tactics[57,58], i.e., individuals may weigh costs and benefits for cooperative decision-making.

Since hierarchies in macaque societies tend to be stable[33,59] (but see ref. 60), the relationships in despotic societies are too, and consequently more so than those within egalitarian societies. Therefore, selective strong bonds may enhance dyadic tolerance, ultimately facilitating (more) discriminate cooperation. These findings, together with the evidence of relatively higher asymmetric and lower transitive grooming interactions, emphasize the role of strong dyadic interdependencies promoting cooperation in more despotic societies. While we provide evidence of these proximate mechanisms underlying cooperation, the ecology of different macaque species may also play influential roles. The extent of intra- and inter-group conflicts and particularly feeding ecologies vary among macaque societies[20,21,30,31], leading to differential needs and interdependencies for cooperative interactions.

Interdependence – a key factor that shapes social tolerance – may vary across group- and dyadic levels[19]. The emergence of dyadic cooperation was best predicted by dyadic social tolerance, even when controlling for the different species and groups. This aligns with the assumption of the interdependency hypothesis that strong dyadic interdependencies can result in higher dyadic tolerance, subsequently leading to higher cooperation[36,37,61–63]. We also found the effects of prosocial motivations and kinship on the likelihood of cooperation, conforming with the findings of previous studies[7,36,64,65]. Despite steeper hierarchies in more despotic societies, discriminate cooperation can be evolutionarily maintained through direct reciprocity and kinship (see discussion[13]). Notably, kinship only explained a small part of the data variance in comparison to reciprocity. Individuals in more despotic societies, thus, can adopt behavioral strategies, such as selective prosocial helping and cooperation[12,13], as rank-related fitness benefits are not always evident[66], and those strategies can potentially aid during active rank mobility[60].

In humans, hierarchies are known to influence cooperation, contingent on how they are operationalized. A large dyadic rank difference can be detrimental to both human and non-human primate cooperation[12,35,67]. However, the 'functionalist' theories suggest that organizational level hierarchy promotes human cooperation through efficient decision-making, motivation, and coordination[68]. Although empirical evidence supporting the functionalist theories is mixed (cf ref. 68), social relationships – through which dyadic interdependence may form – are rarely investigated. Future comparative research incorporating social relationships and dyadic interdependencies may enable us to better understand the functional mechanisms of cooperation. Note that in macaques, despotic and egalitarian social styles are often used synonymously with cooperative and competitive socio-ecologies, respectively. While different strategies have been observed in other non-human primates depending on the socio-ecologies[69,70], in evolutionary agent-based simulations, reciprocity evolved more readily in competitive than cooperative socio-ecologies[71]. Furthermore, within-group conflict is considered to have played a key role in the evolution of cooperation (e.g., parochial altruism[72]), potentially explaining how cooperation can be maintained in more despotic societies, at least selectively among group members. While a larger sample size of groups and careful group-level methodologies would be beneficial in order to test the group-level predictions of the interdependency hypothesis, the current study provides compelling evidence that cooperation can emerge and be maintained even in highly despotic societies through strong dyadic interdependencies.

## Methods
### Study subjects, housing, and husbandry
We studied 13 captive groups of macaques belonging to six different species (*Macaca fuscata*, *M. mulatta*, *M. fascicularis*, *M. silenus*, *M. sylvanus*, and *M. nigra*) in their existing social settings. The *M. fuscata* group was housed at the Affenberg Landskron Park in Villach, Austria, in ca. 40,000 m² enclosure with natural mixed forests typical to

Southern Austria[73]. The population size was ~170. The two *M. mulatta* groups (R3G2 and R3G7) were housed at the Biomedical Primate Research Center (BPRC) in Rijswijk, the Netherlands. The R3G2 and R3G7 groups comprised 30 and 25 individuals, respectively. Both groups had identically designed 74 m² indoor and 250 m² outdoor enclosures. The three *M. fascicularis* groups (J1G4, J1G7, and NWR) were also housed at the BPRC. The group sizes of J1G4, J1G7, and NWR were 15, 18, and 4, respectively. One individual was removed from J1G7 due to within-group compatibility issues before the completion of the studies and, hence, was not included in the analyses. J1G4 and J1G7 had identically sized 49 m² indoor and 183 m² outdoor enclosures. Due to a relatively small group size, the NWR group had 3.55 m² and 3.88 m² indoor and outdoor enclosures, respectively. We studied two *M. silenus* groups, one housed at Blijdorp Zoo and the other at the Apenheul Primate Park, both in the Netherlands. The Blijdorp and Apenheul groups consisted of 3 and 8 individuals, respectively. The Blijdorp *M. silenus* group had 100 m² and 106 m² indoor and outdoor enclosures, respectively, whereas the Apenheul *M. silenus* group had an 80 m² indoor and a 768 m² outdoor enclosure. Of the two *M. sylvanus* groups, one was housed at Gaia Zoo in the Netherlands and the other at the Apenheul Primate Park. The Gaia Zoo and Apenheul *M. sylvanus* groups comprised 14 and 13 individuals, respectively. The Gaia Zoo *M. sylvanus* group had a 108 m² indoor (not for regular use of the animals) and a 3522 m² outdoor enclosure. For the Apenheul *M. sylvanus* group, the only (outdoor) enclosure had an area of 3829 m² with natural vegetation, rocks, and a narrow creek. Finally, the three *M. nigra* groups were housed at Bijdorp, Artis Zoo in the Netherlands, and Planckendael Zoo in Belgium, and the group sizes were 5, 4, and 6, respectively. Blijdorp *M. nigra* group had an indoor enclosure of 70 m² and a 160 m² outdoor enclosure. The Artis *M. nigra* group had 65 m² and 761 m² indoor and outdoor enclosures, respectively. The Planckendael *M. nigra* group had access to only an 82 m² indoor enclosure. For groups with indoor and outdoor enclosures, connecting tunnels enabled macaques to move freely between them (except for the Gaia *M. sylvanus* group). The group sizes reported here included all individuals, including <1 year old. However, for our study, we only considered individuals >1 year at the time of testing (see Supplementary Table S1).

All observational and experimental study procedures were approved by the Animal Experiments Committee and Animal Welfare Organization of BPRC (Animal Welfare Organization/IvD approval no.: 019 A, 019 C, 019D, and 019E). Furthermore, internal committees of all relevant zoos carefully monitored the study. All study components were non-invasive (European Directive 2010/63), and we strictly adhered to the ethical principles and guidelines of the American Society of Primatologists for the care and inclusion of animals. No animals were isolated from their existing social groups during our research.

The husbandry protocols differed slightly across study groups due to species-specific requirements and in-house management decisions, but these protocols adhered to the European Association of Zoos and Aquaria guidelines for accommodation and care for animals[74]. Accordingly, all enclosures had multiple enrichment structures, like climbing platforms, hanging ropes, wooden structures, tree trunks, and slides. Except for the *M. fuscata* and Apenheul *M. sylvanus* groups, which live exclusively in outdoor enclosures, indoor enclosures of all groups were temperature-controlled and had concrete floors covered with sawdust bedding. Depending on the nutritional requirements, feeding routines also varied across groups. The diet primarily consisted of monkey pellets, fresh vegetables and fruits, and seed mix (e.g., sunflower and corn). All study groups had access to drinking water 24/7. No change in the regular feeding schedule was made for our study. The participation of individuals in all of our studies was completely voluntary. Since our study period overlapped with the COVID-19 pandemic years (overall study period: November 2020-February 2023), the zoos and animal holding facilities had strict

preventive measures (e.g., restriction in behavioral experimentation in close proximity to animals) due to macaques' susceptibility to COVID-19. As a result, we could not perform all the tests uniformly across our study groups. In addition, a small subset of data in this study has been used from published research on a single species. See Supplementary Table S1 for details on the observations, different tests, the specific study groups on which we performed them, and the use of published datasets. Multiple experimenters were involved in the collection of data. However, the same experimenter(s) carried out all phases of a single test (cooperation, prosociality, and co-feeding tolerance) for a given study group to avoid potential experimenter bias. We used a randomized order in which behavioral observations, cooperation, and prosociality tests were conducted. The co-feeding tolerance test took place at the end for all study groups. Notably, the study groups were familiarized with the concerned experimenters and showed no distress during the investigation.

## Cooperation test

We used a *loose-string paradigm* to assess the dyadic cooperative tendencies of macaques following a standardized protocol[12]. In this paradigm, the experimental apparatus consisted of a movable platform (width = ca. 1.1 m) placed above a wooden base (width = ca. 1.3 m). Two plastic food trays were installed on either end along the side of the movable platform closest to the macaques. The other side of the platform had a handle attached, which an experimenter could use to move the platform. The food trays were far apart such that an individual could not simultaneously obtain food from both. Two small metal loops were anchored to the platform. Based on the experimental phase, strings were either attached or inserted through the loops, which allowed macaques to move the platform by pulling. As advised by in-house veterinarians, we used unshelled peanuts (halves) and mixed seeds (corn, sunflower, and pea) as food rewards for the cooperation experiment. We found that the participating individuals obtained all food rewards upon successful cooperation trials, suggesting their strong motivation for these food items. The cooperation test consisted of three phases: *habituation, training and social tolerance*, and *testing*, all of which were video recorded using Canon Legria HF G25 and Sony FDR AX100E cameras. During all phases, the identity of the individuals and how often they pulled the strings were recorded.

The habituation phase allowed macaques to familiarize themselves with the experimental apparatus. We placed the apparatus in front of the enclosure so the individuals could interact with it (through touching, licking, and sniffing). Multiple food rewards were placed on the food trays to get the attention of the individuals. The experimenter gently pushed the platform by using the handle for individuals to obtain food rewards. This step was repeated until 50% of the group members obtained at least three food items and became habituated. Habituating half of the *M. fuscata* group was practically challenging. Therefore, we adjusted the criterion for this group from 50% to > 8% (i.e., at least more than 14 individuals) of the population. This percentage was chosen considering the sizes of other relatively large macaque groups in the study. The habituation phase was performed for a maximum of three hours on a single day, for 2-3 days, to fulfill the criterion and proceed to the next phase.

In the training and social tolerance phase, macaques voluntarily self-trained themselves with the pulling mechanism of the apparatus. Two small strings were tightly attached to the metal loops of the moving platform, such that pulling either string by an individual could bring the platform close to them. In addition to self-training, we used this phase to measure dyadic social tolerance among the participating group members. A trial began when the experimenter called "monkeys" and placed two similar food rewards on the two feeding trays. This procedure helped get the attention of the group members. After placing food rewards, the experimenter presented the strings to the macaques. We counted how often macaques obtained food rewards

close to their (specific) group members at the other string. A trial concluded when both food rewards were obtained or after 2 min, whichever was earlier. If macaques did not pull the strings for three consecutive trials, the session was stopped and resumed the next day. A total of 18 sessions were conducted per group, each with 20 trials. Not more than four sessions were conducted on a single day. Approximately 20 s inter-trial and 5-minute inter-session intervals were used. Upon completion of sessions, we investigated whether 50% of each study group and > 8% of the *M. fuscata* group obtained food rewards at least 10 times. All other groups except R3G7 (a rhesus group) fulfilled this criterion. We conducted five additional training sessions with R3G7, but the criterion was still not fulfilled. Nevertheless, we included this group in the next phase of testing.

The testing phase investigated macaques' tendencies to cooperate with group members at the dyadic level. A single and relatively long string was inserted through the metal loops and presented to the macaques. Thus, to move the platform and obtain food rewards, two individuals needed to pull the two loose ends of the string simultaneously. If only one individual pulled, the string unthreaded, and the platform did not move from its original position. A trial began when the experimenter placed food rewards on the trays, called "monkeys", and presented the two loose ends of the string. A trial ended when two individuals simultaneously pulled the strings and successfully moved the platform, or when the string unthreaded due to pulling by one individual. A trial lasted for a maximum duration of 2 min. We conducted a total of 30 sessions per group, with each session having 20 trials. The number of sessions conducted per day and inter-trial and inter-session intervals were identical to that of the training and social tolerance phase.

## Prosociality test

We used a *group service paradigm*[7,44,75,76] to investigate the prosocial motivations of macaques following standardized protocols of seesaw[13] and swing set[77] mechanisms. While the seesaw mechanism was used predominantly in our study, COVID-19 restrictions forced us to use the swing set mechanism for the three *M. fascicularis* groups[77]. This ensured that the macaques were tested from a relatively greater distance, unlike the seesaw mechanism. Notably, the two mechanisms were identical in their representations of the group service paradigm (cf refs. 13,77). We explained the seesaw mechanism below and highlighted its swing set counterparts.

The seesaw mechanism consisted of a wooden board (length = ca. 1.5 m) with two transparent pipes (diameter = ca. 3 inches) attached to two ends (Pos. 0 or provision end and Pos. 1 or receiving end). Food rewards (unshelled peanut, corn seed, sunflower seed, and pea seed) could freely move through the pipes. By default, the wooden board was tilted towards the experimenter's end. A wooden or metal handle was connected to the board next to Pos. 0. The handle was projected towards the macaques' end, which, upon pressing, could tilt the board towards them. When food rewards were present, pushing the handle resulted in items rolling down through the pipes and coming within reach of the macaques. However, an attempt to push or release the handle halfway resulted in the seesaw returning to its initial position, making food out of reach for the macaques. To obtain food rewards, it was thus essential to push the handle fully. Due to the distance between the food pipes, an individual pressing the handle (at Pos. 0) could not reach Pos. 1 to receive food rewards. Thus, macaques could only provision group members. The swing set mechanism used a 1.2 m long wooden beam hung from the ceiling by steel chains. Instead of food pipes, the apparatus included two small plastic buckets (at Pos. 0 and Pos. 1). Unlike the handle in the seesaw apparatus, a rope was attached next to Pos. 0, pulling which could bring the beam with food buckets close to the macaques, and again releasing it, e.g., to try and get access to the other bucket, resulted in the swing moving back to its original position in which the buckets were out of reach.

The group service paradigm consisted of the following phases: *habituation to apparatus, initial training and habituation to procedure, food distribution assessment, apparatus training, group service*, and *blocked control*. The first three phases included general habituation and self-training of the macaques. The prosociality test was video recorded using Canon Legria HF G25 and Sony FDR AX100E cameras. During all phases, the identity of the individuals and the number of times they obtained food rewards were recorded.

In the habituation phase, food items were spread over the wooden board (or placed inside the buckets in the swing set apparatus) at regular intervals for macaques to interact with the apparatus. This phase eliminated any potential effect of neophobia on the experimental setup. On consecutive days, we conducted a total of 2-3 sessions, with each session lasting 60-90 minutes. Individuals were considered habituated when they obtained at least 10 food rewards. Like the loose-string paradigm, we set a target of habituating at least 50% of group members of all study groups, except for the *M. fuscata* group, where this criterion was > 8%. The criterion was fulfilled for all groups.

We decided the number of trials per session for the next phases by considering the sizes of the study groups. The number of trials was n*5 ($n$ = group size with individuals > 1 year) for all groups except *M. fuscata*. In the *M. fuscata* group, we found that 25 individuals were habituated; therefore, the number of trials per session was 25*5 = 125[13].

The seesaw mechanism was fully locked during the initial training and procedure habituation phase, with the wooden platform tilted towards the macaques. Thus, macaques could not push the handle and operate the seesaw. However, upon placing in either position, macaques could retrieve food rewards that rolled down through food pipes. This initial training and habituation phase was not conducted for the swing set mechanism, as individuals were already habituated to the apparatus. Five sessions were conducted, and for each session, food rewards were placed in Pos. 0 and Pos. 1 alternately. A trial started when the experimenter called "monkeys" and placed a food reward in either Pos. 0 or Pos. 1. A trial concluded with an individual obtaining the reward or after a maximum of 2 min. Individuals were considered trained only if they obtained at least 10 food rewards. At least half of the habituated individuals needed to be trained to perform the next phase. This criterion was fulfilled for all groups.

The food distribution assessment investigated group-level social tolerance but only included individuals habituated to the apparatus. The procedure was identical to the previous phase, except the experimenter always placed food rewards in Pos. 1 in this phase. Two sessions were conducted to assess the average within-group food distribution or tolerance.

The seesaw (and swing set) apparatus was unlocked and operational in the apparatus training phase. Thus, macaques could push the handle (or pull the string to move the wooden platform/beam in the swing set). Unlike in the food assessment phase, the experimenter always placed food rewards in Pos. 0, such that the individuals operating the apparatus obtained rewards themselves. At least half of the habituated individuals needed to be trained by obtaining a minimum of 10 food rewards over five sessions. This criterion was fulfilled in all study groups except for two *M. fascicularis* groups (J1G4 and J1G7). Additional sessions (one for J1G4 and two for J1G7) were conducted[77], after which the criterion was fulfilled.

The group service phase consisted of test and empty control sessions. During test sessions, the experimenter placed food rewards in Pos. 1 of the apparatus. We recorded how often individuals pushed the handle (or pulled the string in the swing set) and provisioned food to group members. To control for the potential effect of stimulus enhancement (movement during food placement), we conducted empty control sessions, where the experimenter pretended to place food rewards in Pos. 1. A test trial started with the experimenter calling "monkeys" and placing a food reward in Pos. 1. The trial ended when an individual fully pushed the handle (or fully pulled the string in the swing set) or after 2 min. Note that pushing or pulling did not necessarily lead to food provisioning, as group members needed to be present at the receiving end (i.e., Pos. 1). Similar to the test sessions, an empty control trial began when the experimenter called "monkeys" and pretended to place a reward in Pos. 1, which ended when an individual pushed the handle or after 2 minutes. Five test and five empty control sessions were conducted alternatingly.

We blocked Pos. 1 using a plexiglass in the blocked control phase. This phase assessed whether pushing the handle (or pulling the string in the swing set) was due to self-motivation for food rewards. A trial began when the experimenter called "monkeys" and placed a food reward in Pos. 1. A trial concluded when an individual fully pushed the handle or after 2 min. Therefore, provisioning was impossible even if an individual was present at the receiving end (i.e., Pos. 1). Further, we conducted empty blocked control sessions where the experimenter pretended to place a reward in Pos. 1, similar to the group service phase. Alternatively, five blocked and five empty blocked sessions were conducted. Note that we could not retrieve data from the first three blocked control sessions due to a technical malfunction with the video camera during the investigation of the R3G7 group (a rhesus group). However, to analyze whether individuals understood the paradigm and showed prosocial motivations, only the final two sessions (i.e., sessions 4 and 5) of each condition were considered (cf refs. [7],[13],[44],[75],[76]).

To keep the provisioning macaques motivated, all sessions of the group service and blocked control phases had motivational trials. During these trials, food rewards were placed in Pos. 0. Thus, provisioning individuals could obtain rewards by pushing the handle (or pulling the string in the swing set). Each session began with a motivational trial, repeated after every fifth regular trial. We used 10-second inter-trial and 5 min inter-session intervals throughout the group service experiment. If macaques did not participate in three consecutive trials during a given session in a given phase (except empty and blocked control phases), the session was called off and resumed the next day. Besides, if 'untrained' individuals interacted with the apparatus during the group service or blocked control phase, we waited for them to move away, and repeated the affected trials. Notably, trials were conducted only when at least two trained individuals were present within a radius of 5 m of the experimental setup.

## Co-feeding tolerance test

Based on the group sizes (considering individuals > 1 year), we created testing zones or plots in macaques' enclosures. The width of the plot was always 75 cm, and the length depended on the group size ((group size – 1) x 12.5 cm). For example, in a group of 5 individuals over one year, the plot length was 12.5 × 4 = 50 cm. We uniformly distributed a predetermined number of unshelled and unsalted peanuts (group size x 12 pieces) within the plot. The fundamental assumption was that, in a 'tolerant' group, all individuals would remain in close proximity to each other in a zone full of valuable resources. The proportion of a group present in the plot was used as a proxy for co-feeding tolerance, with a higher proportion suggesting greater tolerance.

The co-feeding tolerance test was conducted primarily in the indoor enclosures of the study groups. However, the test was performed in outdoor enclosures for the two *M. sylvanus* groups and one *M. silenus* group (in Apenheul) due to logistic advantages. Note that the *M. sylvanus* group in Apenheul had only a usable outdoor enclosure. We chose strategic areas in the enclosures that are typically used for feeding. These areas were chosen to make the plots, eliminating the bias of 'not finding' them by macaques. We made visible boundaries in the outdoor enclosures, and the designated areas were cleared of sawdust bedding for indoor enclosures, making them clearly visible. Groups were temporarily locked in their outdoor enclosures (indoor for Gaia *M. sylvanus* group and Apenheul *M. silenus* group). The

experimenter prepared the plot in the designated areas using measuring tapes and uniformly distributed the predetermined number of peanuts. Our study groups receive enrichment food after general lock-unlock procedures (e.g., during enclosure cleaning) as part of standard husbandry protocols. Therefore, we expected the macaques to be motivated to visit the indoor enclosures (and not remain in the outdoor enclosures). The experimental design differed slightly for the Apenheul *M. sylvanus* group. As per veterinary advice, we used hazelnuts instead of peanuts throughout all the sessions. Besides, since this group had only an outdoor enclosure, we did not have the opportunity to lock them elsewhere temporarily. Instead, after preparing the plot, hazelnuts were quickly distributed out of sight of the macaques, followed by catching their attention.

A total of eight sessions were conducted in each study group, except for the *M. nigra* group housed at Planckendael, where we could conduct seven sessions due to veterinary regulations. We controlled for this variation in our statistical analysis. Not more than one session was carried out on a given day. To avoid any potential bias of hunger, co-feeding test sessions took place at least one hour after the regular feeding schedule. A session started when an individual (or individuals) entered the plot and concluded when all the peanuts were eaten. We recorded the activities of the macaques for an additional 2 min after the plot became empty. The co-feeding tolerance test was video recorded using Canon Legria HF G25 and Sony FDR AX100E cameras mounted on tripods.

## Behavioral data collection using continuous focal sampling

Behavioral observations were conducted using continuous focal sampling methods. We conducted 20 min long focal sessions, and after correcting for the time out of sight, we obtained >29,800 minutes of behavioral data (Supplementary Table S1). The observation minutes per group varied (cf. Supplementary Table S1). The focal observational data were used to investigate the dominance-rank relationships, as well as direct affiliative and aggressive interactions. These measures were calculated at the group as well as dyadic levels. Focal data on the *M. fuscata* group could not be collected (cf. Supplementary Table S1), but other ongoing and published research on the *M. fuscata* group provided us with information on dominance-rank relationships[13,73]. Complete data on individual kin relationships were available for all study groups.

We utilized five behavioral variables representing unprovoked submissive behaviors (*avoid*, *be displaced*, *silent-bared teeth*, *flee*, and *social presence*) to construct within-group dominance hierarchies and to assess dyadic dominance-rank relationships[78]. We recorded the frequency of grooming given and received to assess direct affiliative interactions. In addition, we collected data on proximity (within one body length) and affiliative body contact, which were used to validate the grooming data. Finally, we used seven behavioral variables representing contact- and non-contact aggression (*hit*, *chase*, *lunge*, *bite*, *grab*, *open-mouth threat*, and *stare*) to determine the aggressive interactions[79].

## Data coding and statistics

All data were coded from the video recordings using BORIS (version 7.13.6)[80] or in a frame-by-frame manner using Pot Player (version 240618). Due to data coding by multiple coders, we comprehensively checked for inter-rater reliability using intra-class correlation (ICC) tests among the trained coders. Overall agreement was high, with the ICC (3,k) ranging between 0.88 and 0.97. Statistical analyses were performed using R (version 4.4.1)[81]. The agent-based EMO-model was constructed using NetLogo (version 5.3.1).

We applied a Bayesian mixed modeling approach to analyze our empirical data using the *brms* package of R[82]. We used weakly informative priors to guide the estimation process while allowing the data to dominate the posterior distributions[83,84]. A normal prior with a mean

(m) of 0 and a standard deviation (SD) of 5 was used for the intercept term. We applied normal priors for all other fixed-effect coefficients with m = 0 and SD = 1. An exponential prior with a rate parameter of 1 was assigned to the standard deviation of the random effects. Unlike *p*-values in the frequentist approach, we reported median estimate coefficients (*Est*), 89% credible interval (*crI*) that contains 89% of the posterior probability density function, and probability of direction (*pd*), indicating the direction and certainty of an effect. We also verified the results by checking 95% *crI* values and found no differences in comparison to the 89% *crI* ones. Model convergence was assessed following Bayesian statistics guidelines[85]. We investigated trace and autocorrelation plots, Gelman-Rubin convergence estimations, and density histograms of posterior distributions of all the models. We sampled posterior distributions using 10,000 iterations and 2000 warmups.

To account for the hierarchical nature of the dataset and avoid pseudoreplication, all appropriate statistical models included random intercepts for individuals (nested within groups) and for groups nested within species. Specifically, we used the structure (1 | species/group) + (1 | group:ind1) + (1 | group:ind2) to model the dependency among observations. This ensured that group-level observations were not treated as statistically independent species-level replicates. Tolerance grade was included as a fixed effect at the species level, which varied across species. This structure partitioned variance across species, groups, and individuals, allowing us to retain within-species variation without inflating the degrees of freedom for species-level predictors.

## Cooperation

We defined cooperation (or cooperation success) as two individuals in the testing phase simultaneously pulling the strings and obtaining rewards without monopolization. Our analyses focused only on the participating individuals (i.e., at least one pull) who were fully habituated and trained following the different criteria. The percentage of participating individuals varied across groups. We found extremely low participation in R3G7 (one of the *M.mulatta* groups). Notably, this is the same group in which we conducted additional training sessions. We therefore decided to discard the data on this group from the cooperation analyses. The range of participation rate was 35.71% to 100% (mean = 66.94%, standard deviation = 20.05%), excluding the semi free-ranging *M. fuscata* group, where the participation rate was 9.41%.

Cooperation success was coded as a binary variable at the trial level. We counted how often (i.e., in how many trials) two individuals successfully cooperated in a group (see Supplementary Data S1 for the list of trained individuals). The total number of trials in the testing phase for each group was 600. However, the removal of an individual from J1G7 (one of the *M. fascicularis* groups) led us to adjust the number of trials in which the individual participated (cf ref. 12). This resulted in an adjusted 459 testing trials in J1G7. Accordingly, we incorporated an offset term in our statistical models to control the varying numbers of testing trials. To assess whether macaques generally understood the need for partners to obtain food rewards, we compared the instances when they pulled strings alone vs. in the presence of partners. We found that macaques pulled the strings more often with a partner than when present alone (Wilcoxon signed rank test: $Z = -2.18$, Cohen's $d = 0.59$, 95% CI [0.06, 1.08], $P = 0.028$). Although delayed control trials may provide a better understanding of the cognitive processes underlying the need for cooperation partners[86], our measure in a social experimental setting potentially indicated the same (also see ref. 12). Moreover, we were interested in investigating the spontaneous collaborative efforts of two individuals rather than the cognitive mechanisms facilitating active partner recruitment.

Data from the social tolerance phase of the cooperation test was used to measure dyadic tolerance levels. The social tolerance phase allowed individuals to co-feed or monopolize the food rewards. We followed a standardized method, where all co-feeding and monopolization opportunities were considered[12]. The number of times two specific individuals obtained food by sitting in proximity (i.e., two attached strings at the apparatus) was divided by the sum of these individuals feeding alone and monopolizing. The total number of trials in the social tolerance phase was 360 for each group, except for J1G7. Due to the removal of an individual in J1G7, similar to the testing trials, the number of social tolerance trials was adjusted. Instead of 360, it was 192 for J1G7. For each study group, dyadic social tolerance values were standardized, with higher values indicating higher dyadic tolerance.

We investigated the effects of eight social and demographic dyadic attributes or predictors – prosocial motivations, dyadic social tolerance, kinship, rank- and age differences, sex compositions, and dyadic affiliative and aggressive relationships on cooperation. Empirical evidence suggests that dyads composed of at least one individual with prosocial motivation can lead to higher cooperation success than dyads consisting of individuals without any prosocial motivations[36]. Thus, we called a dyad 'prosocial' when at least one of the members was identified to have some prosocial motivation (Supplementary Data S2). For the dyadic social tolerance measure, we analyzed data obtained from the dedicated social tolerance phase of the loose-string paradigm (cf. Fig. 1B)[12]. Dyadic social tolerance was measured by the tendency of individuals to obtain food rewards, with and without monopolization, in the presence of group members. We used a genetic relatedness cutoff of 0.25 to determine whether a dyad was kin-related. When calculating the within-group steepness of hierarchies[87], we determined the dominance-rank relationships, particularly the ordinal rank differences among individuals. Focal data were used to create dyadic grooming and aggressive indices[88].

We used two sets (each encompassing eight groups) of Bayesian mixed models (i.e., hurdle models) for the analyses as data on socio-demographic factors based on behavioral observations were not uniformly available (cf. Supplementary Table S1). The first set of models investigated the likelihood and magnitude of cooperation with all dyadic social and demographic attributes, except for grooming and aggression indices. These two indices were used as predictors in a second set of models. The relevant models included rank difference and grooming index as interaction terms with the tolerance grades. Interaction terms were dropped in case no robust effect was found in subsequent simpler models. Individuals nested within groups and groups nested within species were included as random effects in the models.

## Prosociality

Participation (i.e., with complete habituation and training) in the group service paradigm was slightly higher than in the cooperative loose-string paradigm. It ranged between 61.50% and 100% (mean = 81.48%, standard deviation = 12.98%), excluding the *M. fuscata* group, where the rate was 14.70% (cf ref. 13). Subsequently, instead of using the percentage of food provisioning (cf refs. 7,75), we reported the number of individuals with prosocial motivations in a group as a measure of group-level prosociality.

We used three criteria to determine whether an individual had prosocial motivation. First, an individual needed to push the handle (or pull the rope in swing set apparatus[77]) more often in group service test than at least one of the control conditions (i.e., empty control and blocked control). This criterion was set because the number of test trials in which individuals could push the handle was restricted by the number of participants and the number of pushes by other individuals. Therefore, pushing significantly more in the test than in both empty and blocked control conditions was a strict and conservative cut-off.

Second, although pushing the handle and subsequently provisioning food may require spatiotemporal coordination among group members, i.e., the presence of a receiver at Pos. 1, an individual was only considered to have prosocial motivations when food was provisioned to a group member. Third, an individual proactively provisioned food to group members, i.e., without relying on solicitation from the receiver or showing aggression towards the receiver[13,75,77]. We used Fisher's exact tests to investigate whether an individual pushed the handle more in the test than at least one of the control conditions. Notably, we only used data from sessions 4 and 5 (of group service test, empty control, and blocked control) for analyses as macaques needed to learn the contingencies of the control conditions[7,13,44,76,77]. The number of individuals with prosocial motivations ranged between 0 and 9 across groups (mean = 2.33, standard deviation = 2.59, Supplementary Data S2). All individuals, other than the ones with prosocial motivations (i.e., with experimental evidence), were considered 'non-prosocial'. This was decided as all group members had the opportunity to participate and provide food (if prosocial) to each other in the group service paradigm.

The food distribution assessment phase provided information on group-level tolerance but was restricted to only individuals who interacted with the apparatus. We calculated Pielou's evenness index (or Pielou's *J*)[42]. The index can range between 0 and 1, with higher values (i.e., closer to 1) indicating more uniform food distribution among group members. In our study groups, Pielou's *J* ranged between 0.13 to 0.65 (mean = 0.41, standard deviation = 0.19, Supplementary Data S4).

## Co-feeding tolerance

For estimating group-level co-feeding tolerance, we calculated cumulative presence, defined by the proportion of individuals in the group present both inside the plot and within one arm's length of reach of the plot at regular time intervals. To eliminate any potential effects of varying enclosure sizes and distances between unlocking zones (if applicable) and the plot, counting started (i.e., first scan) when an individual first entered the plot. Scans were conducted at each 10 s interval until the plot became empty. The number of scans per session per group varied greatly (range = 2 – 27, mean = 8.15, standard deviation = 4.91), primarily depending on the level of participation and monopolization rate. That is, more monopolization led to longer feeding duration, resulting in a higher number of scans. By investigating the obtained mean value, we decided on a cut-off of eight scans consistently across all sessions for all groups. Thus, data collected up to eight scans were utilized for the co-feeding tolerance analyses. During statistical analysis for the co-feeding tolerance, we also checked the results with an even smaller cut-off of six scans for validation. Macaque tolerance grades (i.e., grades 1 to 4) were included as an interaction term with scans. We constructed an intercept-only Bayesian mixed model to obtain the median co-feeding tolerance values of the groups. The median co-feeding tolerance ranged between 0.15 and 0.81, with higher values suggesting a higher co-feeding tolerance. Additionally, statistical models were built to compare macaque societies along the despotic-egalitarian gradient, where the cumulative presence variable was used as a function of the categorical tolerance grades and scans.

## Dominance-rank relationships

We calculated group-level hierarchy steepness and determined the ordinal ranks of individuals along the corresponding hierarchies. A Bayesian Elo-rating method was used for these calculations[87]. Based on unprovoked submissive behaviors, we prepared directional matrices. The Bayesian Elo-rating method calculates winning probabilities from these matrices to assess the steepness of the hierarchy. A steepness range of 0.23 to 0.91 was found (mean = 0.65, standard deviation = 0.22, Supplementary Data S4), with higher values indicating more

steeper hierarchies. For each group, dyadic rank differences were calculated and standardized for statistical analysis. As the macaque co-variation framework is built upon interactions among adult females, we also calculated within-group hierarchy steepness based on only adult females. Using these data, we found a steepness range of 0.24 to 0.93 (mean = 0.61, standard deviation = 0.24, Supplementary Data S4).

## Affiliative and aggressive interactions

We carried out social network analyses to measure within-group affiliative interactions. Specifically, we calculated global transitivity, reciprocity, and modularity values from grooming matrices (Supplementary Data S4). In more egalitarian macaque societies, grooming distribution is less likely to be asymmetric, as less steep hierarchies in egalitarian societies do not hinder grooming from being symmetrically present[23,25,26]. In contrast, in more despotic societies, most grooming interactions are concentrated among a few group members (typically including the dominant individuals or kin relatives), making the grooming networks modular and asymmetric. Within-group grooming transitivity values ranged between 0.24 and 1 (mean = 0.76, standard deviation = 0.26), with more transitive networks indicating a symmetrical distribution of grooming among group members. Reciprocity ranged between 0.27 to 1 (mean = 0.75, standard deviation = 0.19), with higher values suggesting more symmetrical or reciprocal grooming patterns. Finally, modularity had a range of 0 to 0.33 (mean = 0.76, standard deviation = 0.26), with higher values indicating the presence of more distinct modular grooming networks.

We calculated the group-level frequencies of aggressive interactions. All instances of aggressive behaviors were combined and corrected for the group's overall observation minutes to obtain within-group aggression values. Notably, tolerance grades did not correlate with the frequency of aggression per minute ($r = 0.14$, $n = 12$, 89% $crl = [-0.37, 0.59]$, Fig. 2B), suggesting that the frequency of aggression was comparable across the despotic-egalitarian gradient.

In addition to group-level measures, we calculated dyadic affiliative and aggressive indices using standardized methods[12,88]. The observed frequencies of grooming, proximity, body contact, and aggression were extracted separately for all combinations of dyads. These values were then divided by the combined observation minutes of the individuals making the dyad. The resulting rates were further divided by a group average of the corresponding behavior (i.e., grooming, proximity, and body contact). Finally, we z-transformed the values to control for group-level variance. We specifically used grooming data for our analyses. To see whether grooming effectively captured affiliative relationships, we carried out Bayesian correlation tests between dyadic grooming and dyadic proximity ($r = 0.45$, $n = 203$, 89% $crl = [0.34, 0.55]$, $pd = 1$), and dyadic grooming and affiliative body contact ($r = 0.10$, $n = 203$, 89% $crl = [-0.04, 0.22]$, $pd = 0.92$). These results indicate that dyadic affiliative interactions, to a certain extent, were positively reflected by grooming.

## Agent-based EMO-model

The agent-based EMO-model simulates macaque social behavior, where all 'agents' or individuals share the same behavioral rules but have variable internal states[50–52,54]. These rules and internal states determine how actors interact, further influencing their internal states. Besides moving around in space, agents can engage in affiliative (grooming and affiliative signaling) and agonistic (attacking, aggressive, or submissive signaling) behavior. The EMO-model has been validated by comparing its results to empirical data on various species of free-living macaques[52].

The agents in the EMO-model are capable of emotional book-keeping, integrating information of past affiliative interactions into a partner-specific LIKE attitude, and establishing a LIKE relationship. More frequent affiliative interactions (i.e., receiving grooming from a group member) increase the LIKE attitude associated with the group member, positively affecting the likelihood of affiliative social interactions with this group member[52]. Reciprocal grooming relationships emerge in the model when grooming and LIKE within a dyad are mutually reinforcing. However, this only occurs in a limited part of the parameter space, provided that actors are very selective when choosing a partner[50] and that LIKE attitudes decrease over time with intermediate speed[51]. However, the decreased speed that leads to the most naturalistic scenario, in turn, depends on the speed with which LIKE increases during grooming and the dynamics with which it increases and decreases[54]. In contrast to a LIKE attitude, every agent also assigned a FEAR attitude to every other agent. The FEAR attitude does not change over time or due to social interactions; thus, it models a static dominance hierarchy. For instance, the most dominant individual (individual A) has a FEAR value of 1/20, while the least dominant individual has a FEAR value of 20/20. FEAR-attitudes (from individual A towards individual B) are calculated by subtracting individual B's FEAR value from individual A's FEAR value (in this example: 0.05-1 = −0.95). FEAR attitudes, therefore, range from −0.95 (A is dominant over B) to 0.95 (B is dominant over A).

In order to manipulate the steepness of the dominance hierarchy, we added a variable to the EMO-model called *Hierarchy steepness*. The Hierarchy steepness variable ranged from 0.2 to 1, with higher values indicating more despotic societies ('1' = despotic, '0.6' = intermediate, and '0.2' = egalitarian). This variable scaled the range of FEAR attitudes in a group. For example, in a simulation run with a Hierarchy steepness setting of 0.1, FEAR attitudes would only range from −0.095 to 0.095, making the hierarchical (or rank) difference between the most dominant and least dominant individual smaller than 0.1. Since FEAR attitudes only impact the likelihood of aggressive and submissive behavior, the manipulation of hierarchy steepness only directly impacts the likelihood of aggression and submission. In contrast, the likelihood of affiliative behavior was only affected indirectly. To study the impact of Hierarchy steepness on the distribution of social relationships (as measured by LIKE values), we performed every simulation run three times, with each repeat having a different setting of Hierarchy steepness. The effect of Hierarchy steepness was investigated for two different dynamics settings (the original or easy-going, following[50–52], as well as the alternative or picky, following[54]), three different increase speeds (fast, intermediate, and slow; cf ref. 54) and three different decrease speeds (720, 2880 and 5400, cf rfs. [51,54]). The easy-going LIKE dynamics settings assume that increasing LIKE follows a linear curve while decreasing LIKE follows an exponential curve. In contrast, LIKE increases and decreases following a logistic curve while using the picky dynamics settings. As a result, when using the picky dynamics, it is more difficult to form new strong relationships, but once a strong relationship is established, its quality decreases more slowly over time, compared to a relationship of the same quality in a simulation run using the easy-going dynamics setting. All simulation runs were performed with very high partner selectivity (0.99; cf ref. 50).

We investigated the impact of Hierarchy steepness on the distributions of LIKE values (averaged over the 'second year' of the recording period). Level plots were made for each simulation run, showing the LIKE attitude of agents toward each other (resulting in 20 × 20 level plots). Level plots were visually inspected. To quantify the stability of LIKE over time, LIKE distributions were measured at five different points in time, separated by 0.25 years (start of the second year of the recording period, after 1.25, 1.5, 1.75 years, and at the end of the recording period). For stability assessment, an $R^2$ was calculated for each transition (i.e., between LIKE distributions of consecutive time points). The resulting four $R^2$ values were averaged, resulting in an average 'stability' score for each simulation run (see refs. 50–52,54 for details). Notably, all group-level properties in the EMO-model were emergent properties determined by the interactions among the agents.

Every unique run (i.e., every set of parameter settings tested) was repeated three times. These runs with the same parameter settings, but using a different random seed, always resulted in very similar stability scores. This was true for the entire parameter space except for runs with picky dynamics, a slow increase speed, and a fast decrease speed. For these runs, a fourth repeat was performed, to get a better understanding of what the more likely outcome would be. Although it still generally holds true for these settings that a steeper hierarchy will restrict the LIKE distribution, resulting in a more stable LIKE distribution, this is not true for every repeat. Because these runs have a slow increase speed and a fast decrease speed, combined with picky dynamics, it is very difficult to form a high-LIKE relationship, especially with a very steep hierarchy. This results in a scenario with only 1 or 2 high-LIKE dyads that remain stable over time, resulting in a high stability score. However, in some instances, there are no dyads that managed to form a high-LIKE relationship, resulting in a scenario without any stable high-LIKE relationships and a correspondingly low stability score (of 0). When hierarchy strength is low enough (Hierarchy strength = 0.2), it seems slightly easier to form somewhat stable relationships. While these runs can also be very stable, similar to the more despotic runs, no stability scores under 0.3 were observed.

### Reporting summary

Further information on research design is available in the Nature Portfolio Reporting Summary linked to this article.

## Data availability

All data generated in this study are available within the article, its supplementary information, and on Zenodo https://doi.org/10.5281/zenodo.18366379. There are no restrictions on data availability. Source data are provided in this paper.

## Code availability

All code and accessory files to reproduce results are uploaded on Zenodo https://doi.org/10.5281/zenodo.18366379.

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

## Acknowledgements

This work was supported by a grant from the European Union's Horizon 2020 – Marie Skłodowska-Curie Actions research and innovation program under the grant number H2020-MSCA-IF-2019-893016 (to D.B.). We thank all the staff members of the Affenberg Zoobetriebsgesellschaft mbH (Austria), the Biomedical Primate Research Centre (the Netherlands), Diergaarde Blijdorp (the Netherlands), Gaia Zoo (the Netherlands), Apenheul Primate Park (the Netherlands), Planckendael Zoo (Belgium), and Artis Zoo (the Netherlands) for their support and assistance during the research work. We sincerely thank Peter Gaubatz, Svenja Gaubatz, Max Dorner, Jan A.M. Langermans, Annet Louise Louwerse, Jos Hartog, Linda Bruins-van Sonsbeek, Emile F. Prins, Lisette van den Berg, and Marjolein Osieck for allowing us to conduct the studies at the different zoos and institutions. We thank Sjoerd Sijbrandij, Paola Meems, Amber Kozanli, Yesper Bos, and Mary Maximiadi for their help with data organization. We thank Sudeshna Chakraborty, Julia Ostner, Oliver Schülke, and Thomas Bugnyar for their feedback on an earlier version of the manuscript.

## Author contributions

D.B. and J.J.M.M. conceptualized the study; D.B., T.W.Z., E.J.C.vL., and J.J.M.M. developed the methodology; D.B., T.S.R., and T.W.Z. analyzed the data with input from E.J.C.vL. and J.J.M.M.; D.B., T.S.R., T.W.Z., and V.S. designed and edited the figures with input from J.J.M.M.; D.B., E.B., S.C., P.E.C., E.C., J.A.dJ., E.J.A.M.dL., A.R.G., E.J.J., C.E.K., P.E.N.K., E.M., V.I.S., E.S.J.vD., J.V., S.W., and A.Z. collected and coded the data. D.B. and J.J.M.M. secured funding for this work; D.B. wrote the original manuscript with input from J.J.M.M. D.B., T.W.Z., T.S.R., K.R.L.J., L.S.P., E.H.M.S., E.J.C.vL., and J.J.M.M. edited the manuscript. All authors read and approved the manuscript.

## Funding

## Competing interests

The authors declare no competing interests.

## Additional information

**Debottam Bhattacharjee** [1,2,3] ✉, **Tonko W. Zijlstra** [4], **Tom S. Roth** [1], **Elena Belli** [5,6], **Sophie Calis** [1], **Paula Escriche Chova** [1], **Eythan Cousin** [1,7], **Jolanda A. de Jong** [8], **Edwin J. A. M. de Laat** [1], **Aníta Rut Guðjónsdóttir** [1], **Karline R. L. Janmaat** [4,9,10], **Elja J. Jeunink** [9], **Charlotte E. Kluiver** [1], **Penny E. N. Kuijer** [11], **Esmee Middelburg** [1], **Lena S. Pflüger** [12], **Veera I. Schroderus** [13,14], **Eva S. J. van Dijk** [1], **Jonas Verspeek** [15,16], **Sophie Waasdorp** [1,17], **Adam N. Zeeman** [9,10], **Elisabeth H. M. Sterck** [1,17], **Edwin J. C. van Leeuwen** [1,18] ✉ & **Jorg J. M. Massen** [1,19] ✉

[1]Animal Behaviour & Cognition, Department of Biology, Utrecht University, Utrecht, The Netherlands. [2]Department of Infectious Diseases and Public Health, Jockey Club College of Veterinary Medicine and Life Sciences, City University of Hong Kong, Kowloon, Hong Kong SAR. [3]Centre for Animal Health and Welfare, Jockey Club College of Veterinary Medicine and Life Sciences, City University of Hong Kong, Kowloon, Hong Kong SAR. [4]Department of Cognitive Psychology, Faculty of Social Sciences, Leiden University, Leiden, The Netherlands. [5]Institute of Evolutionary Anthropology, University of Zurich, Zürich, Switzerland. [6]Department of Physics, Chemistry and Biology, IFM Biology, Linköping University, Linköping, Sweden. [7]Department of Ecology, Physiology & Ethology, Faculty of Life Sciences, University of Strasbourg, Strasbourg, France. [8]Department of Applied Biology, Aeres University of Applied Sciences, Almere, The Netherlands. [9]Institute of Biodiversity and Ecosystem Dynamics, Department of Evolutionary and Population Biology, University of

Amsterdam, Amsterdam, The Netherlands. ¹⁰ARTIS Amsterdam Royal Zoo, Amsterdam, The Netherlands. ¹¹Behavioural Ecology Group, Wageningen University and Research, Wageningen, The Netherlands. ¹²Department of Behavioral & Cognitive Biology, University of Vienna, Vienna, Austria. ¹³Department of Biological and Environmental Science, University of Jyväskylä, Jyväskylän yliopisto, Finland. ¹⁴Department of Biology, University of Turku, Turku, Turun Yliopisto, Finland. ¹⁵Behavioural Ecology and Ecophysiology, University of Antwerp, Wilrijk, Belgium. ¹⁶Antwerp Zoo Centre for Research and Conservation, Royal Zoological Society of Antwerp, Antwerp, Belgium. ¹⁷Animal Science Department, Biomedical Primate Research Centre, Rijswijk, The Netherlands. ¹⁸Department for Comparative Cultural Psychology, Max Planck Institute for Evolutionary Anthropology, Leipzig, Germany. ¹⁹Royal Rotterdam Zoological & Botanical Gardens, Rotterdam, The Netherlands. ✉e-mail: bhattacharjee.debottam@gmail.com; edwin_van_leeuwen@eva.mpg.de; j.j.m.massen@uu.nl

