## [Transparent Peer Review file · Nature Communications]

Despotism promotes dyadic cooperation through enhanced interdependencies in non-human primate societies

Corresponding Author: Dr Debottam Bhattacharjee

Version 0:

Reviewer comments:

Reviewer #1

(Remarks to the Author)

This paper is based on an extraordinary data set comparing the behavioral responses of six macaque species to three experimental setups and an agent-based model. The results are clear and without any doubt need to be published. My concern is the introduction of the introduced interdependency hypothesis, which easily can be mistaken by the interdependence hypothesis. There are three different levels of cooperation: dyadic cooperation, group-level cooperation and inter-group cooperation. What the authors discuss here is dyadic cooperation. The authors look at the interdependencies of prosocial behavior, cooperation and tolerance on a dyadic level.

A large body of research has formulated the exchange of commodities in primates and other mammals looking at processes of reciprocity (e.g. Jaeggi & van Schaik 2011, de Waal 1989, Mitani 2006, de Waal & Brosnan 2006). To me it remains unclear, what the interdependency hypothesis explains that for dyadic cooperation reciprocity cannot explain. Of course, the authors can argue that reciprocity is due to proximate mechanisms that follow interdependencies.

I am not convinced yet that the interdependency hypothesis provides anything new or can explain anything better than our model of reciprocal exchange of commodities. But I am more than happy to be convinced.

I want to state this here clearly again: this paper has a great data set and a great potential. It shows without doubt that despotic societies do exchange more cooperation, potentially because for them dyadic cooperation provides more benefit than for more egalitarian societies. The analysis and the results are clean and need absolutely to be published. This study provides a crucial data set and results for our understanding of dyadic cooperation. Whether we need the interdependency hypothesis for this remains unclear to me.

Jaeggi, Adrian V., and Carel P. Van Schaik. "The evolution of food sharing in primates." *Behavioral Ecology and Sociobiology* 65.11 (2011): 2125-2140.

De Waal, Frans BM. "Food sharing and reciprocal obligations among chimpanzees." *Journal of Human Evolution* 18.5 (1989): 433-459.

Mitani, John C. "Reciprocal exchange in chimpanzees and other primates." *Cooperation in primates and humans: mechanisms and evolution*. Berlin, Heidelberg: Springer Berlin Heidelberg, 2006. 107-119.

De Waal, Frans BM, and Sarah F. Brosnan. "Simple and complex reciprocity in primates." *Cooperation in primates and humans: Mechanisms and evolution*. Berlin, Heidelberg: Springer Berlin Heidelberg, 2006. 85-105.

(Remarks on code availability)

Reviewer #4

(Remarks to the Author)

After reading the revised manuscript and the authors' responses to the reviewers' comments, I find that I still share the same concerns raised by the previous reviewers, some of which, unfortunately, have not been fully addressed. Below, I outline only the two most important points.

The authors rely solely on the egalitarian–despotic gradient proposed in a previous study and classify the six species

accordingly, without providing empirical data from the present study to support this categorization. This is problematic because substantial variation is known to exist even within species, and social structure can be strongly influenced by a small number of individuals, group composition, or the physical environment in captivity. Consequently, it is very difficult to interpret the results in terms of group-level characteristics based on these metrics alone.

The mechanisms underlying dyadic cooperation and group cooperation can differ substantially, particularly in terms of their payoff structures. As the authors note, there may be a link between these two forms of cooperation; however, this relationship remains highly controversial. More importantly, the present study does not provide empirical evidence that bears directly on this link. Although the experiments are primarily designed to examine dyadic cooperation, the discussion extends far beyond this scope, which seems excessive for the type of empirical data presented.

Overall, I suggest that the authors revise the manuscript to focus more concisely on dyadic cooperation rather than group cooperation. If it is not possible to include empirical measures of group-level metrics in the present study, this limitation should be stated more explicitly and discussed more clearly as a disadvantage of the study.

(Remarks on code availability)

Response to Reviewers comments:

[Line numbers correspond to the revised version of the manuscript without tracked changes]

We sincerely thank the editor for considering a revised version of the manuscript for publication in *Nature Communications*. As suggested by the editor and the reviewers, we have made substantial changes to the manuscript. Please see the point-by-point response below and the revised manuscript.

Reviewer #1 (Remarks to the Author):

This paper is based on an extraordinary data set comparing the behavioral responses of six macaque species to three experimental setups and an agent-based model. The results are clear and without any doubt need to be published. My concern is the introduction of the introduced interdependency hypothesis, which easily can be mistaken by the interdependence hypothesis. There are three different levels of cooperation: dyadic cooperation, group-level cooperation and inter-group cooperation. What the authors discuss here is dyadic cooperation. The authors look at the interdependencies of prosocial behavior, cooperation and tolerance on a dyadic level.

We sincerely thank the reviewer for their positive response. We have now emphasized our focus on dyadic cooperation in the revised manuscript.

While we focused on and tested the interdependency hypothesis with regard to its predictions on dyadic cooperation, the hypothesis also potentially explains group-level cooperation, which was not explicitly tested (at least at the operational level) in the current study. Thus, the fundamental difference between the interdependence and interdependency hypothesis is that the latter provides explanations to both group-level and dyadic-level cooperation.

To avoid any confusion between the two hypotheses, we have added “cf. interdependence hypothesis” in line 83 to make it explicit. Please also see lines 75 – 92 where we explicitly stated the three levels of cooperation and how our interdependency hypothesis can explain cooperation (at dyadic and group levels).

Lines 75 – 92:

“Growing empirical research also shows that individuals in several species that are neither self-domesticated nor cooperatively breeding exhibit prosocial and cooperative tendencies, primarily through reciprocity⁸⁻¹⁶. While reciprocity is important for various levels of cooperation^{2,17,18}, such as dyadic, group, and inter-group, it can be considered a positive behavioral expression of the structural interdependencies¹⁹, defined by an individual’s stake in another²⁰. However, empirical evidence on how interdependencies may form, function, and eventually lead to cooperation, remains limited. Building on the idea that interdependence has a considerable role in the evolution of human cooperation (cf. interdependence hypothesis)^{20,21}, an overarching interdependency hypothesis may explain what might cause tolerant and less tolerant species, and others in general, to show both group- and dyadic-level cooperative tendencies¹³. This hypothesis suggests that interdependencies at the group level, for example, strength in numbers in colonially nesting species or allomaternal care in cooperatively breeding species, can lead to enhanced within-group social tolerance and promote indiscriminate prosociality and cooperation. It further emphasizes that at the dyadic

level, preferential strong associations or friendships, nepotistic biases, and reliance on coalitions may result in selectively enhanced tolerance, promoting discriminate prosociality and cooperation, particularly in less tolerant or despotic societies ^{12,13}.”

Also, see lines 110 – 113 for the explicit prediction of our hypothesis regarding dyadic cooperation.

Lines 110 – 113: “Such advantageous yet selective preferences among both kin- and non-kin group members can indicate the presence of stronger dyadic interdependencies in despotic than egalitarian societies, leading to enhanced tolerance and discriminate prosocial- and cooperative- tendencies ¹³.”

A large body of research has formulated the exchange of commodities in primates and other mammals looking at processes of reciprocity (e.g. Jaeggi & van Schaik 2011, de Waal 1989, Mitani 2006, de Waal & Brosnan 2006). To me it remains unclear, what the interdependency hypothesis explains that for dyadic cooperation reciprocity cannot explain. Of course, the authors can argue that reciprocity is due to proximate mechanisms that follow interdependencies.

I am not convinced yet that the interdependency hypothesis provides anything new or can explain anything better than our model of reciprocal exchange of commodities. But I am more than happy to be convinced. I want to state this here clearly again: this paper has a great data set and a great potential. It shows without doubt that despotic societies do exchange more cooperation, potentially because for them dyadic cooperation provides more benefit than for more egalitarian societies. The analysis and the results are clean and need absolutely to be published. This study provides a crucial data set and results for our understanding of dyadic cooperation. Whether we need the interdependency hypothesis for this remains unclear to me.

Jaeggi, Adrian V., and Carel P. Van Schaik. "The evolution of food sharing in primates." *Behavioral Ecology and Sociobiology* 65.11 (2011): 2125-2140.

De Waal, Frans BM. "Food sharing and reciprocal obligations among chimpanzees." *Journal of Human Evolution* 18.5 (1989): 433-459.

Mitani, John C. "Reciprocal exchange in chimpanzees and other primates." *Cooperation in primates and humans: mechanisms and evolution*. Berlin, Heidelberg: Springer Berlin Heidelberg, 2006. 107-119.

De Waal, Frans BM, and Sarah F. Brosnan. "Simple and complex reciprocity in primates." *Cooperation in primates and humans: Mechanisms and evolution*. Berlin, Heidelberg: Springer Berlin Heidelberg, 2006. 85-105.

We thank the reviewer for this comment. While we focused on and tested the interdependency hypothesis with regard to its predictions on dyadic cooperation, the hypothesis also potentially explains group-level cooperation, which was not explicitly tested (at least at the operational level) in the current study. Reciprocity is indeed crucial, however it can be considered as a behavioral expression of positive interdependencies at various levels – dyadic, group, and intergroup (see De Dreu et al. 2024). Thus, the fundamental difference between the interdependence and interdependency hypothesis is that the latter provides explanations to both group-level and dyadic-level cooperation.

Ref: De Dreu, C. K., Gross, J., & Romano, A. (2024). Group formation and the evolution of human social organization. *Perspectives on Psychological Science*, 19(2), 320-334.

Regarding the exchange of commodities, we indeed emphasize the proximate mechanisms and motivations in addition to kin selection. Please note that food sharing is not commonly observed among macaques. However, our experimental approach provides evidence of food sharing through the group service and cooperative loose-string paradigms. Furthermore, as kinship only explained some variance in our data for dyadic cooperation, the proximate mechanisms became necessary to focus on. Crucially, we did not find any empirical evidence for an effect of grooming on cooperation, yet in (more) despotic societies, reciprocal exchange of grooming (see Fig 4E) was relatively less. This suggests that exchange of commodities may not always be seen from grooming interactions, and that they can be present in other forms, such as support during conflicts, and broadly ‘Machiavellian’ tactics. The interdependency hypothesis emphasizes these explanations for selectively enhanced tolerance and cooperative interactions.

We have included the suggested references in the revised manuscript. Please see lines 237 – 244:

“Accordingly, these findings opened up the idea that more despotic societies possibly restrict the number of strong (cooperative) bonds based on affiliative interactions (e.g., grooming, also see ^{16,18,46} for food sharing) and that they may be used as a commodity traded in exchange for alternate services, like support during conflicts (*sensu* biological market theory) ⁴⁷. As food sharing is not common in macaque societies, we constructed agent-based models simulating macaque social behavior primarily based on grooming interactions and emotional bookkeeping (see e.g., ⁴⁸) along a despotic-egalitarian gradient.”

Reviewer #4 (Remarks to the Author):

After reading the revised manuscript and the authors’ responses to the reviewers’ comments, I find that I still share the same concerns raised by the previous reviewers, some of which, unfortunately, have not been fully addressed. Below, I outline only the two most important points.

We sincerely thank the reviewer for their feedback on our manuscript. We have addressed the remaining comments.

The authors rely solely on the egalitarian–despotic gradient proposed in a previous study and classify the six species accordingly, without providing empirical data from the present study to support this categorization. This is problematic because substantial variation is known to exist even within species, and social structure can be strongly influenced by a small number of individuals, group composition, or the physical environment in captivity. Consequently, it is very difficult to interpret the results in terms of group-level characteristics based on these metrics alone.

The egalitarian–despotic gradient in macaques is a well defined and established framework. Please see some references below –

Thierry, B. Unity in diversity: Lessons from macaque societies. *Evol. Anthropol.* 16, 224–238 (2007).

Thierry, B. Covariation of conflict management patterns across macaque species. *Natural Conflict Resolution* (2000).

Duboscq, J. *et al.* Social tolerance in wild female crested macaques (*Macaca nigra*) in Tangkoko-Batuangus Nature Reserve, Sulawesi, Indonesia. *Am. J. Primatol.* 75, 361–375 (2013).

Thierry, B. Where do we stand with the covariation framework in primate societies? *American Journal of Biological Anthropology* 178, 5–25 (2022).

Sueur, C. *et al.* A comparative network analysis of social style in macaques. *Anim. Behav.* 82, 845–852 (2011).

We respectfully disagree with the comment that no empirical data were provided in our work to define this gradient. Please see lines 136 – 146 and 225 – 237, and also Fig. 4E.

Lines 136 – 146: “...we first tested whether the macaque co-variation framework assumption regarding steepness of hierarchies (i.e., steeper hierarchies in more despotic societies) is met (cf. Fig 1A).

A negative but weak correlation was found between tolerance grades and hierarchy steepness ($r = -0.33$, $n = 12$, 89% $\text{crl} = [-0.70, 0.19]$, Data S4, Table S2), indicating a trend that despotic societies indeed have steeper hierarchies than egalitarian ones. However, the assumption of the co-variation framework is built upon within-group interactions among adult females^{22,23,25}. Accordingly, based on only adult females in the groups (applicable to 10 groups), we re-calculated steepness and re-investigated the relationship. In these data, a stronger negative correlation was found ($r = -0.61$, $n = 10$, 89% $\text{crl} = [-0.87, -0.11]$, Data S4, Table S2), reliably indicating that the assumption was clearly met.”

Lines 225 – 237: “Based on the frequency of grooming, we built social networks and calculated global transitivity, reciprocity, and modularity. Transitivity provides information on the level of clustering and is particularly useful for interpreting weighted and directed social networks^{26,45}. Reciprocity, by contrast, is a measure showing the difference between grooming efforts given and received. Modularity denotes how a network can be divided into communities where individuals groom specific partners more frequently than chance level²⁷. We found both transitivity and reciprocity to be positively correlated with macaque tolerance grades (Transitivity: $r = 0.73$, $n = 12$, 89% $\text{crl} = [0.38, 0.90]$; Reciprocity: $r = 0.70$, $n = 12$, 89% $\text{crl} = [0.33, 0.89]$, Fig. 4E, Table S2), indicating the presence of more transitive and reciprocal grooming networks in more egalitarian societies. Further, more egalitarian societies showed a lower tendency of grooming modularity than more despotic societies ($r = -0.36$, $n = 12$, 89% $\text{crl} = [-0.72, 0.16]$, Table S2). The observed grooming patterns thus aligned with the assumptions of the co-variation framework^{22,23,25}.”

We completely agree with the reviewer that substantial variation may exist at the intra- and inter-species levels. We also found evidence for the same in our co-feeding tolerance measure and highlighted that. Please see lines 160 – 161 and Fig. 2B.

Lines 160 – 161: “Substantial intra- and inter-species variation in co-feeding tolerance was found (Fig. 2B).”

The mechanisms underlying dyadic cooperation and group cooperation can differ substantially, particularly in terms of their payoff structures. As the authors note, there may be a link between these two forms of cooperation; however, this relationship remains highly controversial. More importantly, the present study does not provide empirical evidence that bears directly on this link. Although the experiments are primarily designed to examine dyadic cooperation, the discussion extends far beyond this scope, which seems excessive for the type of empirical data presented.

We agree with the reviewer comment. Accordingly, we have now emphasized our focus on dyadic cooperation in the revised manuscript. However, while we focused on and tested the interdependency hypothesis with regard to its predictions on dyadic cooperation, the hypothesis also potentially explains group-level cooperation, which was not explicitly tested (at least at the operational level) in the current study. Please see lines 67 – 96 and 110 – 113 in the introduction.

Overall, I suggest that the authors revise the manuscript to focus more concisely on dyadic cooperation rather than group cooperation. If it is not possible to include empirical measures of group-level metrics in the present study, this limitation should be stated more explicitly and discussed more clearly as a disadvantage of the study.

We agree with the reviewer comment and now have emphasized our focus on dyadic cooperation in the revised manuscript (see lines 67 – 96 and 110 – 113 in the introduction). Regarding the empirical evidence of group-level metrics (on grooming, hierarchy steepness, and co-feeding tolerance), please see our response above. Nonetheless, we agree that a larger sample size of groups and careful group-level methodologies would be beneficial to test the group-level predictions of the interdependency hypothesis. We have added this information in the discussion now. Please see lines 342 – 346:

Lines 342 – 346: “While a larger sample size of groups and careful group-level methodologies would be beneficial to test the group-level predictions of the interdependency hypothesis, the current study provides compelling evidence that cooperation can emerge and be sustained even in highly despotic societies through strong dyadic interdependencies.”